# Fourier ring correlation simplifies image restoration in fluorescence microscopy

Sami Koho [1,2], Giorgio Tortarolo[1,3], Marco Castello[1], Takahiro Deguchi[4], Alberto Diaspro [4,5] & Giuseppe Vicidomini [1]

Fourier ring correlation (FRC) has recently gained popularity among fluorescence micro-scopists as a straightforward and objective method to measure the effective image resolution. While the knowledge of the numeric resolution value is helpful in e.g., interpreting imaging results, much more practical use can be made of FRC analysis—in this article we propose blind image restoration methods enabled by it. We apply FRC to perform image de-noising by frequency domain filtering. We propose novel blind linear and non-linear image deconvolution methods that use FRC to estimate the effective point-spread-function, directly from the images. We show how FRC can be used as a powerful metric to observe the progress of iterative deconvolution. We also address two important limitations in FRC that may be of more general interest: how to make FRC work with single images (within certain practical limits) and with three-dimensional images with highly anisotropic resolution.

[1] Molecular Microscopy and Spectroscopy, Istituto Italiano di Tecnologia, Genoa 16152, Italy. [2] Department of Cell Biology and Anatomy, Laboratory of Biophysics, Institute of Biomedicine and Medicity Research Laboratories, University of Turku, Turku 20520, Finland. [3] Dipartimento di Informatiche, Bioingegneria, Robotica e Ingegneri dei Sistemi, University of Genoa, Genoa 16145, Italy. [4] Nanoscopy and NIC@IIT, Istituto Italiano di Tecnologia, Genoa 16152, Italy. [5] Dipartimento di Fisica, University of Genoa, Genoa 16146, Italy. Correspondence and requests for materials should be addressed to S.K. (email: sami.koho@iit.it) or to G.V. (email: giuseppe.vicidomini@iit.it)

Reliable and realistic estimation of spatial resolution in fluorescence microscopy images has for decades been a subject of passionate scientific debate. With the development of fluorescence nanoscopy techniques[1] this debate has resurfaced with new-found fervor, e.g., see refs. [2,3].

Typically resolution is estimated by measuring either the minimum resolvable distance between two adjacent structures in an image—as per the classical Rayleigh/Abbe/Sparrow resolution definitions—or, alternatively it can be estimated from the intensity profile analysis, such as simple full-width-half-maximum (FWHM) or more complex fitting[4] of subresolved sized structures. In order to perform either one of the two measurements, suitable structures need to be subjectively identified and manually measured; ideally, the measurements should be repeated at several positions to gain some statistical basis for the estimate. This task is both tedious and error-prone.

Fourier ring correlation (FRC)[5,6] and Fourier shell correlation (FSC)[7]—essentially, FRC generalized to 3D—have for decades been used to estimate image resolution in electron cryomicroscopy. Recently FRC was adapted for optical nanoscopy, by us[8] and others[9,10], to address the issues with the traditional resolution assessment methods. It is based on a normalized cross-correlation histogram measure calculated in the frequency domain between two images of the same region-of-interest, with independent noise realizations. For FRC/FSC calculation, the spatial frequency spectra of the two images are divided into bins, which produces a series of concentric rings/shells in the polar-form frequency domain images (hence the names). The FRC/FSC histogram is formed by calculating a correlation value for each bin according to

$$\mathrm{FRC/FSC}_{12}(r_i) = \frac{\sum_{r \in r_i} F_1(r) \cdot F_2(r)^*}{\sqrt{\sum_{r \in r_i} F_1^2(r) \cdot \sum_{r \in r_i} F_2^2(r)}} \qquad (1)$$

where $F_1$ and $F_2$ are the Fourier transforms of the two images and $r_i$ the $i$th frequency bin. The image resolution in FRC/FSC is defined from the histogram, as a cut off frequency at which the cross-correlation value drops below a preset threshold value. Advantages of the FRC/FSC are that it is fully automatic, quantitative, and depends both on sample and microscope characteristics. It is also less prone to subjective bias and measurement errors, although the choice of the appropriate resolution threshold criterion still requires some input from the researcher, as no single solution seems to be correct in all applications[9,11].

While the resolution estimation certainly is an interesting application in itself, in our view, the true potential of FRC/FSC is in much more practical tasks. In this paper we show several examples of advanced image restoration methods that leverage FRC/FSC measures. We apply FRC to perform image denoising by frequency domain filtering. We propose novel blind linear and iterative image deconvolution methods that use FRC/FSC measurements to estimate the effective point-spread-function (PSF) of the microscope, directly from the images, with no need for prior knowledge of the instrument characteristics. The deconvolution is shown to work exquisitely with both two- and three-dimensional (2D/3D) images. We also show how FRC can be used as a powerful metric to observe the progress of iterative deconvolution tasks.

## Results

### The role of FRC in image restoration.
In image restoration tasks, in one way or another, one tries to improve the quality of an image, by increasing the contrast between signal and noise, i.e., the signal-to-noise ratio (SNR). The FRC measure defines a cut off frequency in the frequency domain, beyond which (at higher frequencies than the cutoff), there are no details with sufficient SNR to be discernible. The knowledge of the cut off frequency alone (i.e., resolution in the spatial domain), can already help in several image restoration and analysis tasks, such as frequency domain filtering, autofocusing and image quality assessment. On the other hand, it is a common practice to interrelate the spatial resolution with the FWHM of the microscope's PSF. Thus, once one knows the resolution (FRC), it is possible to form an estimate of the PSF. Naturally, an assumption needs to be made regarding the shape of the PSF—here, we approximate the PSF as a simple Gaussian function, which should be reasonable in most cases[12]. This makes it possible to implement blind image deconvolution (deblurring) algorithms that leverage FRC to obtain an estimate of the PSF. In iterative deconvolution tasks it may also be of interest to update the PSF after each iteration, or to evaluate the quality of the deconvolution results. The principle of using FRC in image restoration and tasks is illustrated in Fig. 1a. The list of tasks mentioned in the Fig. 1a is by no means intended to be exhaustive, but it just simply reflects subjects that we touch upon in this paper.

In Fig. 1b,c an example is shown about the use of FRC in Fourier domain low-pass filtering with a noisy confocal image of HeLa cell tubulin cytoskeleton (pixel size 51 nm). The result shown in Fig. 1c was obtained with an ideal low-pass filter, which simply removes all the frequencies that are higher than then FRC cutoff (183 nm$^{-1}$). The ideal filter works excellently: it is able to practically remove all the high-frequency noise, with a faint low-frequency noise pattern visible in the background (from scanning, laser fluctuation etc.). The use of an ideal filter is not typically recommended, as introducing such sharp edges into the frequency domain may produce artefacts (ringing effects) in the filtering results. No such effects can be seen here, supposedly because signal power is very low at frequencies higher than the FRC threshold. In Supplementary Fig. 1 results obtained with three different Fourier domain filters ideal, Butterworth and Gaussian are compared. In addition a result for the ideal filter with a theoretical cut off value is shown. All the filters were able to significantly reduce the noise, with no apparent effect on the fine image details, but the ideal filter with FRC cutoff is clearly the most effective.

In Fig. 1d an example is shown, of 2D blind Wiener deconvolution, with FRC-based PSF. The PSF is based on a simple Gaussian model, in which the FRC resolution is used as FWHM value (FWHM = $2\sqrt{2\ln 2}\sigma$). The Wiener deconvolution result shows dramatic enhancement of contrast and apparent resolution, but Wiener filtering does not really have any effect on the background level. It is possible to further tune the performance of a Wiener filter with the regularization parameter value. In Fig. 1d rather gentle regularization (SNR = 0.1) was used to produce the crisp looking result; the value was chosen subjectively.

### FRC enabled blind iterative deconvolution.
For traditional FRC analysis two images of the very same region-of-interest and with independent noise realizations are needed. There are various ways to obtain these two images, as discussed, e.g., in reference[8]. In iterative image deconvolution, however, especially if one wants to explore the feedback connection illustrated in Fig. 1a—for progress estimation or iterative PSF updates—the two-image requirement is not optimal, as it would require simultaneously, in a synchronous manner, running deconvolution on two images. This translates into complex software implementation, a significant computational overhead, and possible hardware resource issues, especially when dealing with large images. And of course, one does not always have two identical images in the first place.

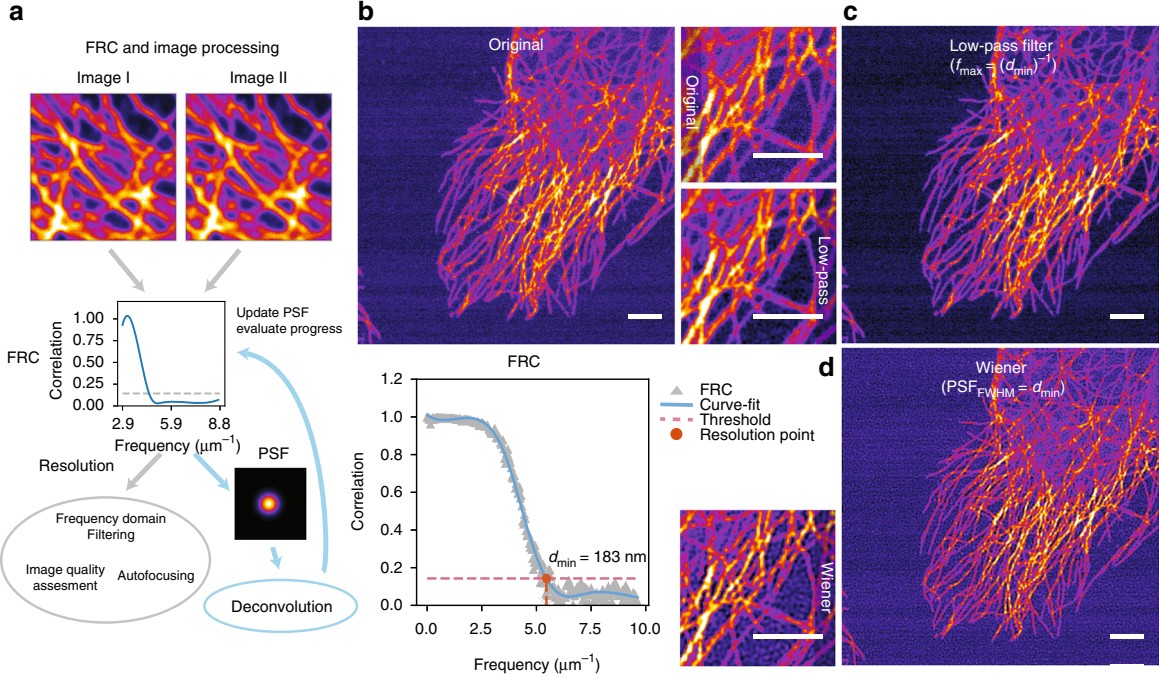

**Fig. 1** Application of FRC to image processing. **a** Various ways to leverage FRC measures in image processing tasks are illustrated. **b** FRC measure on a HeLa cell image is shown to yield a resolution $d_{min}$ of 183 nm. **c** $d_{min}^{-1}$ is used as a cut off value in a frequency domain filter. **d** The $d_{min}$ value is used as a full-width-half-maximum (FWHM) value for a Gaussian PSF, in a blind Wiener filter. Scale bars 3 μm

For this reason, we wanted to figure out, whether it would be possible to find a method to calculate FRC from a single image (one-image FRC), with reasonable compromises, in terms of frequency bandwidth and accuracy.

In order to perform FRC analysis on a single image, one needs to find a way to form statistically independent image subsets—that share the same details, but different noise realizations—by some form of subsampling. As described in Fig. 2a, we propose to do this by dividing a single image into four subsets, i.e., two-image pairs. The first pair is formed by taking every pixel with (even, even) row/column indexes to form one subimage and (odd, odd) indexes to the other. The second image pair is formed from pixels at (even, odd) and (odd, even) indexes. The dimensions of the four subimages are identical, exactly half of the size of the original image. FRC can be calculated from either one of the image pairs alone, but we noticed that averaging two measurements helps to deal with special spectral domain symmetries (Supplementary Fig. 2) that arise when details in an image are oriented predominantly in one direction. With 3D images (FSC) the same splitting method is used, except that in the axial direction (z), layers are summed pairwise to maintain image proportions; we get back to the 3D measurements later.

Because subsampling inevitably leads to loss of information, one might be keen to think that such a method is only feasible on significantly oversampled data, i.e., with sampling density much higher than the Nyquist limit ($d_{pixel} \leq d_{min}/2\sqrt{2}$; $d_{pixel}$ for pixel size; $d_{min}$ for expected image resolution); we use the $\sqrt{2}$ factor in our Nyquist limit definition to ensure sufficient sampling in all directions, assuming square pixels on a rectangular sampling grid[13]. However, as illustrated by the diagonal lines in Fig. 1a, the proposed subsampling pattern introduces a shift between the two subimages in each of the two-image pairs, which as described in detail in Supplementary Note 1, demonstrates itself as a rather interesting exponential modulation in the frequency domain. This modulation has the property of compressing the FRC curve; by

compression one means that details are shifted to a lower frequency than where they actually should be. In order to make any use of this effect, however, we needed to figure out how the compression works, i.e., where are the frequencies shifted. We achieved this by calibrating the one-image FRC, against the "gold standard" two-image FRC, at the cut off frequency defined by the 1/7 resolution threshold[8,9]; the same threshold was used in both one- and two-image FRC. The principle of the calibration is explained in detail in Supplementary Note 2, and the results are shown in Supplementary Fig. 3. After the calibration, the one- and two-image FRC work in a very similar manner, as shown in Fig. 2b, up to the Nyquist limit. It should be noted that the calibration is only valid for the stated 1/7 threshold; we do not attempt to correct the entire FRC curves. For a different threshold, a similar process should be repeated. This however does not stand in the way of using one-image FRC in the image restoration applications that we envision for it here.

Having established how to calculate FRC on single images, we then put the method to work in iterative Richardson–Lucy (RL) devonvolution. In our RL implementation, the FRC is calculated after each iteration in order to assess the effective resolution of the deconvolution results, similar to what was done in[14]. We implemented the RL algorithm in two ways: in regular RL the same FRC-based PSF is used throughout the iteration process, whereas in the Adjustive RL the PSF is updated after each iteration based on FRC. The latter case was inspired by classical iterative blind deconvolution algorithms[15,16], in which one starts with some sort of a rough PSF estimate that is updated at every iteration. Such a scheme may be beneficial if the initial guess does not fit the data very well. When using FRC for the PSF estimation, the first guess is always based on the actual data, but for example when the original data is very noisy (low SNR), FRC may initially underestimate the resolution. Updating the PSF throughout the iteration process may however lead to overestimation of the PSF, and thus using some sort of a constraint may be beneficial. To

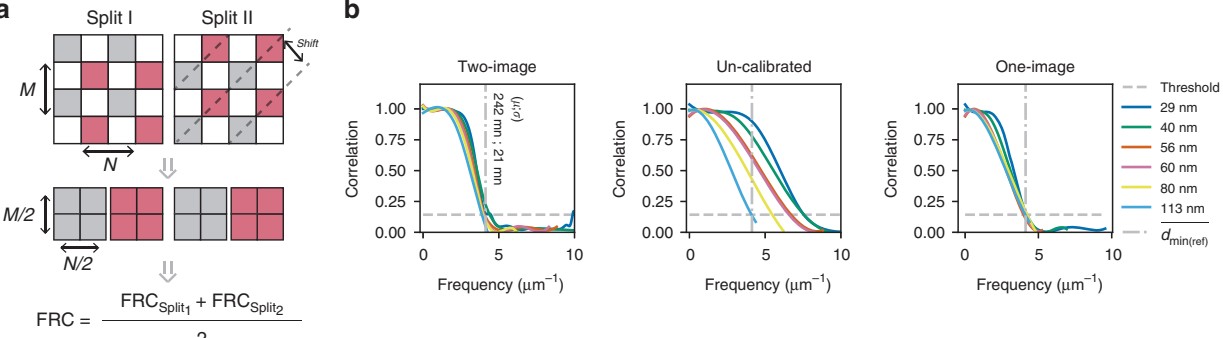

**Fig. 2** Illustrating one-image FRC. **a** The principle of one-image FRC is illustrated. First image is split into two subimage pairs, after which FRC is calculated for both the pairs and then averaged. The purpose of the averaging is to deal with frequency domain asymmetries. The one-image FRC was calibrated against the two-image FRC with a series of confocal images of a cell with vimentin staining. The field of view and focal plane in each image is the same; this made it possible to evaluate FRC measures at a nearly fixed resolution, with different sampling densities (pixel sizes vary from 29 to 113 nm). The calibration results are shown in Supplementary Fig. 3. **b** Calibrated one-image and two-image FRC measures are compared. The uncalibrated curves were plotted assuming the same pixel size with the two-image FRC, hence the apparent shift of the FRC curves to the right; the calibration curve also corrects for the different pixel pitch. The $\overline{d_{\min(\text{ref})}} = 242$ nm; 12 nm ($\mu$; $\sigma$) denotes the average resolution measured with two-image FRC

that end, in Adjustive RL-TV we added total variation (TV) regularization ($\lambda_{\text{TV}} = 5 \times 10^{-4}$)[17] to the Adjustive RL algorithm.

The blind RL deconvolution results with the three methods are shown in Fig. 3a with a microtubules-stained HeLa cell confocal image (pixel size 56 nm). The regular RL deconvolution, with fixed PSF produces the sharpest looking results. The Adjustive RL and Adjustive RL-TV both improve the quality of the original image, but at least in this instance, there seems to be no benefit from updating the PSF during iteration. In Fig. 3b the FRC resolution is plotted as a function of iteration count, for each of the three algorithms. The measures confirm the subjective observations: the regular RL produces the highest resolution results. It is also interesting to observe that the TV regularization helps the Adjustive RL algorithm to converge much more quickly, but the improvement stops at a similar level with the nonregularized Adjustive RL.

From the FRC measures overall, it is evident that the effective resolution in the deconvolution results tends to converge to a nearly fixed value after a number of iterations. This insight motivated us to look into another long-standing problem with RL and other iterative deconvolution algorithms: that one does not really know when the algorithm should be stopped[17–19]. Quite commonly this is decided by trial and error, by running the algorithm with different iteration counts, and picking the best looking result. Several parameters have been proposed as well to observe the progress of deconvolution, but sadly, they are not very reliable or commonly used. Therefore we wanted to see, if we could use FRC for that, and found out that the first derivative ($\nabla$Resolution) of the curve shown in Fig. 3c is very well suited for that—it quantifies the rate of change of the FRC resolution as a function of deconvolution iterations. The most obvious iteration to stop the deconvolution would be at the maximum resolution, i.e., when the derivative reaches zero. In Fig. 3 Adjustive RL converges after 27 iterations, Adjustive RL-TV after 17, and RL after 92 iterations. However, as is evident from the RL deconvolution, reaching the maximum resolution may take a long time, and on the other hand, very little improvement is made (based on the FRC measures) as the deconvolution approaches convergence. Thus, the derivative zero may be an overly stringent condition. Therefore we highlight two alternative thresholds in Fig. 3c ($\nabla$ Resolution $\geq -1$ nm it$^{-1}$, $\nabla$ Resolution $\geq -0.2$ nm it$^{-1}$) that one may want to use instead. The first threshold stops the iteration, when most of the resolution gain has been made, whereas at the second threshold, the deconvolution approaches

nearly complete convergence. The specified resolution thresholds assume a nanometer length scale, which is meaningful in optical microscopy. One of course needs to adjust for appropriate physical units, when working with images with significantly higher or lower resolution. The 1 nm it$^{-1}$ threshold corresponds to roughly 0.5% and 0.02 nm it$^{-1}$ to 1 permille of the resolution. In Supplementary Fig. 4 the deconvolution results with the RL algorithm at the three thresholds are compared ($\nabla$ Resolution $\geq -1$ nm it$^{-1}$, $-0.2$ nm it$^{-1}$, and $0$ nm it$^{-1}$). Already at the lower $-1$ nm it$^{-1}$ threshold the results are very good. In Supplementary Fig. 5 similar results were obtained with a vimentin stained cell image (confocal, pixel size 29 nm) and in Supplementary Fig. 6 with a widefield image of a vimentin stained cell. In all of the three examples (Fig. 3d, e; Supplementary Fig. 5e, f; Supplementary Fig. 6e, f) two alternative deconvolution progress parameters $\tau_1$ and $\eta_k$ are plotted as well: both indicate some sort of convergence, but no other qualitative information can be derived from them. The $\eta_k$ measure in Fig. 3d works exactly the same with all three algorithms; $\tau_1$ reacts to the regularization, but cannot tell a difference between the RL and Adjustive RL algorithms.

Having established that the FRC-based deconvolution progress observation works rather well, we then wanted to test its robustness to strong background noise, which is known to affect the performance of RL (and other iterative deconvolution algorithms)[20]; the algorithm(s) cannot make a difference between signal and background noise, and thus, as the deconvolution progresses, it eventually starts to fit the noise in the background as well as the signal[21,22]. In order to assess how this effect becomes visible in the FRC measures, we devised a worst case simulation of sorts with an image containing mainly background noise (Supplementary Fig. 7). After the very first iteration a separate second peak appears in the FRC curves that grows in intensity as the RL algorithm tries to fit the noise into a sparse solution, which results into the dotty pattern that is shown after 30 iterations. In this case, as shown in Supplementary Fig. 7b,c, the FRC does not get caught into the second peak, but manages to measure the actual resolution, and the deconvolution stops after one iteration. It is however possible that a situation would arise, in which the noise peak is not as clearly separated from the signal, and thus we wanted to devise some sort of an indicator for abnormal behavior of the FRC measures during deconvolution. For this purpose in Supplementary Fig. 7d, we propose the derivative of the $\nabla$ Resolution curve $\nabla^2$ Resolution. The

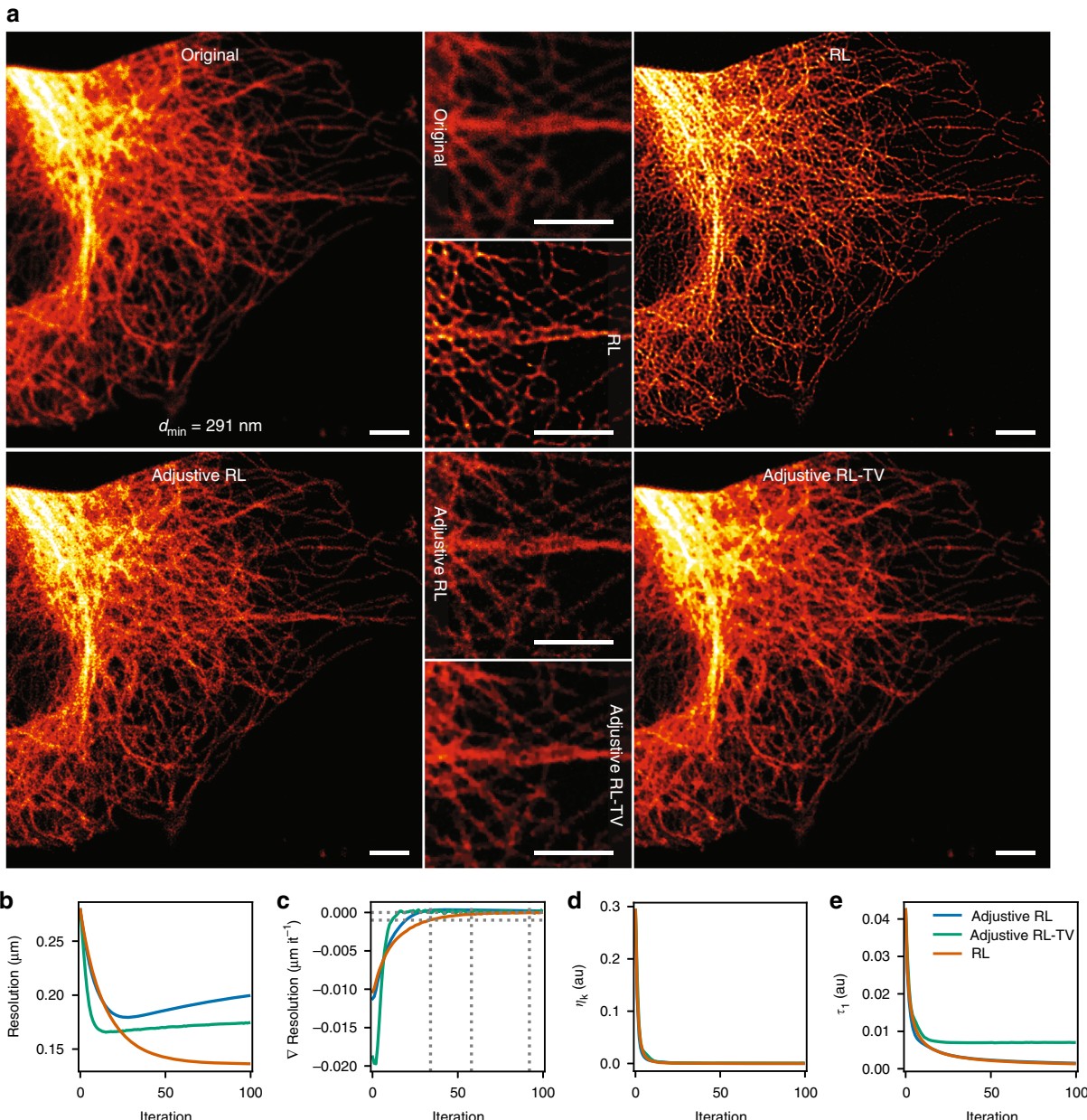

**Fig. 3** Blind iterative RL deconvolution enabled by FRC measurements. **a** Deconvolution results of a confocal image of a microtubules-stained HeLa cell are shown, with three variants of the RL algorithm. In RL the same FRC-based Gaussian PSF (FWHM = 291 nm) is used throughout the iteration, whereas in Adjustive RL the initial PSF estimate is updated after every iteration. In Adjustive RL-TV in addition total variation regularization is used ($\lambda_{TV} = 5 \times 10^{-4}$). The deconvolution of each algorithm was stopped at the iteration of maximal FRC resolution, which is plotted in **b** as a function of the iteration count (RL 92, Adjustive RL 27, and Adjustive RL-TV 15 iterations). **c** The first derivative of **b** is shown. Iterations at which the derivative ∇ Resolution (i.e., $\nabla d_{min}$) drops below predefined thresholds −1 nm it$^{-1}$, −0.2 nm it$^{-1}$, and 0 nm it$^{-1}$ (maximum resolution) in the RL algorithm are highlighted with dashed lines. The RL results at these thresholds are compared in Supplementary Fig. 4. **d**, **e** The $\tau_1$ and $\eta_k$ curves are shown. Scale bars 3 μm

reasoning behind this is that as the deconvolution converges, the slope of the ∇ Resolution should approach zero; i.e., it should not significantly increase, after it has decreased previously. The $\nabla^2$ Resolution crosses zero every time ∇ Resolution changes direction, thus indicating abnormal convergence behavior—it works as a failsafe of sorts.

In Supplementary Fig. 8 an RL deconvolution example of a mictotubulin-labeled HeLa cell with strong noise background is shown. It is possible to improve the performance of the RL algorithm with images with strong background, as discussed in ref. [20], by adding an estimate of the background to the algorithm

as a prior information. To that end in Supplementary Fig. 8, a (blind) method for estimating the background is introduced, and it is shown to greatly enhance the quality of the deconvolution result. As shown in Supplementary Fig. 8b the strong background demonstrates itself as a shoulder in the FRC curves, which appears after seven iterations. Unlike in the simulation, the background does not form a separate peak, but rather gets mixed with the signal spectrum. The FRC curves with the background estimation do not show a similar effect, and the RL deconvolution also seems to reach a much higher resolution (degree of convergence). This is confirmed by FRC measurements, shown

in Supplementary Fig. 8d. The appearance of the shoulder in the FRC curves shows up as an edge in the $\nabla$ Resolution curve (Supplementary Fig. 8d), and as a zero crossing $\nabla^2$ Resolution (Supplementary Fig. 8e), which is in agreement with the simulations. The $\eta_k$ (Supplementary Fig. 8f) and $\tau_1$ (Supplementary Fig. 8g) do not react to the background fitting in any way.

**Working with 3D images.** In fluorescence microscopy the image resolution is highly anisotropic: due to limited numerical aperture of the single objective lens microscope systems, the axial resolution (direction of the optical axis) is typically at least factor of three inferior to the lateral resolution. For this reason FSC, the simple 3D expansion of FRC, is of only limited value in fluorescence microscopy applications. In order to address this issue, we developed a FSC-based method in which each Fourier Shell is divided into wedges. A single Sectioned FSC (SFSC) indexing shape consists of two such wedges that are each others mirror images (Supplementary Fig. 9b)—this is to take advantage of the symmetries in Fourier space. In order to calculate resolution values for the whole sphere, the dual-wedge structure is rotated in increments of $\alpha$ (Supplementary Fig. 9a) around an axis located on the XY plane—for each orientation a separate cross-correlation histogram and resolution value are calculated; the $\alpha$ also matches the angular size of the wedge. At $\alpha = 2\pi$ the SFSC simplifies into normal FSC. As a concept the SFSC is very similar to the recently proposed Conical FSC measure[23]. They essentially differ in the way that the Fourier sphere is indexed to produce the directional resolution measures. The SFSC is especially tuned to observe variation of resolution, when rotating around a single axis, which makes it rather fast to calculate (few sections/volume) as well as robust (large number of voxels on every section).

We compared our new SFSC measure with FRC on a stimulated emission depletion (STED) microscopy[24] image stack of a microtubules stained HeLa cell. The FRC measures were made by identifying apparently in-focus planes in the 3D images and then calculating separate resolution values for the XY and XZ orientations. We also compared our SFSC measure against Fourier Plane Correlation (FPC) that was proposed for 3D image analysis in[9]. In order to facilitate the manual FRC measurement and to limit the computational load in SFSC and FPC the resolution measurements were done on a cropped $300 \times 300 \times 30$ pixel central section of the STED image (about 1/3 of the image size, $300 \times 300 \times 300$ pixels after resampling and zero padding). As shown in Supplementary Fig. 10, the FRC and SFSC measurements are in rather good agreement. SFSC measurement at $SNR_e = 0.5$ threshold (one-bit[11]), corresponds to FRC measurement at 1/7 threshold; the one-bit threshold was used in all SFSC measurements. The FPC works rather well as well, but appears to suffer from the interpolation, scanning and other artefacts in the axial direction, which as shown in Supplementary Fig. 10a, c causes it to somewhat underestimate the resolution in the axial direction.

In Fig. 4, blind Wiener filtering results with two 3D images are shown. The new SFSC measure is leveraged to produce the resolution estimates, necessary for generating the PSFs. In Fig. 4a, results obtained with a STED microscope super-resolution image are shown. The images were acquired with relatively low STED depletion intensity to ensure good contrast, and to reduce photobleaching. As is typical to a STED microscope image, the resolution anisotropy between lateral (XY) and axial (Z) directions is considerable, as shown in the SFSC resolution plot in Fig. 4a. The polar plot illustrates the resolution point in micrometers, and it was calculated by rotating the SFSC section at 15° increments around the $y$-axis. Because STED is a bandwidth unlimited technique, the power spectrum in STED images often

contains very high frequencies that unfortunately, are often hidden by noise. However, the simple blind Wiener filtering, as shown in Fig. 4a, is able to recover a surprisingly large amount of fine details. The axial haze is clearly reduced, and the effective resolution is drastically improved with previously blurred filaments, clearly visible in the results.

In Fig. 4b, the same blind Wiener filtering approach was applied to a much larger (deep) image of Pollen recorded with a confocal microscope. Only single image was available for analysis, so the diagonal splitting was used in the SFSC calculations. The deconvolution results show dramatic improvement of contrast and details—and axial haze is effectively reduced, as indicated by the depth coloring. Even crispier details can be obtained by decreasing the Wiener regularization; SNR = 0.005 was used in Fig. 4b to produce a smooth result.

**Discussion**

In this paper several blind image restoration methods were introduced that leverage FRC resolution measurements in different ways. In frequency domain denoising methods FRC was used to find a cut off frequency point for low-pass filtering. In deconvolution tasks, both linear (Wiener) and iterative (RL), FRC measurements were used to estimate the effective PSF, directly from the image data. There are several clear benefits of estimating the PSF with FRC. Firstly, no prior knowledge—even theoretical—is needed of the microscope or the sample. Secondly, the PSF generated via FRC is always tailored to every given image. Thirdly, the PSF estimation with FRC is a single-step process, although it can also be updated iteratively if necessary. This advantage specifically made it possible to perform linear blind Wiener filtering in a very straightforward way, both in 2D and 3D. With larger images it might be of interest to divide the images into several smaller blocks[25], to adapt the PSF in the blind deconvolution to local changes of resolution (Supplementary Fig. 11).

In iterative deconvolution (RL), FRC measures were also leveraged to observe the progress and quality of the deconvolution. No other such metric to our knowledge exists in the literature. The $\tau_1$ and $\eta_k$, as well as many other measures that can be found in the literature, mainly quantify the convergence of the deconvolution algorithm, but cannot really quantitatively analyze the quality of the deconvolution results, in absence of a ground truth image. It was shown that FRC can be used to identify the deconvolution iteration at which the effective resolution is maximal—or near to it, if using the proposed thresholds. It was also shown that FRC measures can provide additional qualitative information of the deconvolution progress, e.g., related to the effect of the image background to the deconvolution quality.

In addition to deconvolution and image denoising that was the focus of this paper, there are several other image processing/analysis tasks that FRC/SFSC could be applied to. In Supplementary Fig. 12, we entertain the idea of combining FRC with other image quality assessment parameters, to produce quantitative measures of image quality for e.g., high-content screening applications; a similar method was recently proposed for assessing the quality of localization super-resolution microscopy image reconstructions[26]. FRC is also very sensitive to the de-focus: as shown in Supplementary Fig. 13, it actually behaves rather similarly to several autofocus metrics, which might be an interesting future application—and of course, a good way to identify an in-focus plane at postprocessing stage. Our open-source MIPLIB software library that was used to perform all the demonstrated image analysis and processing tasks, may help in such future applications.

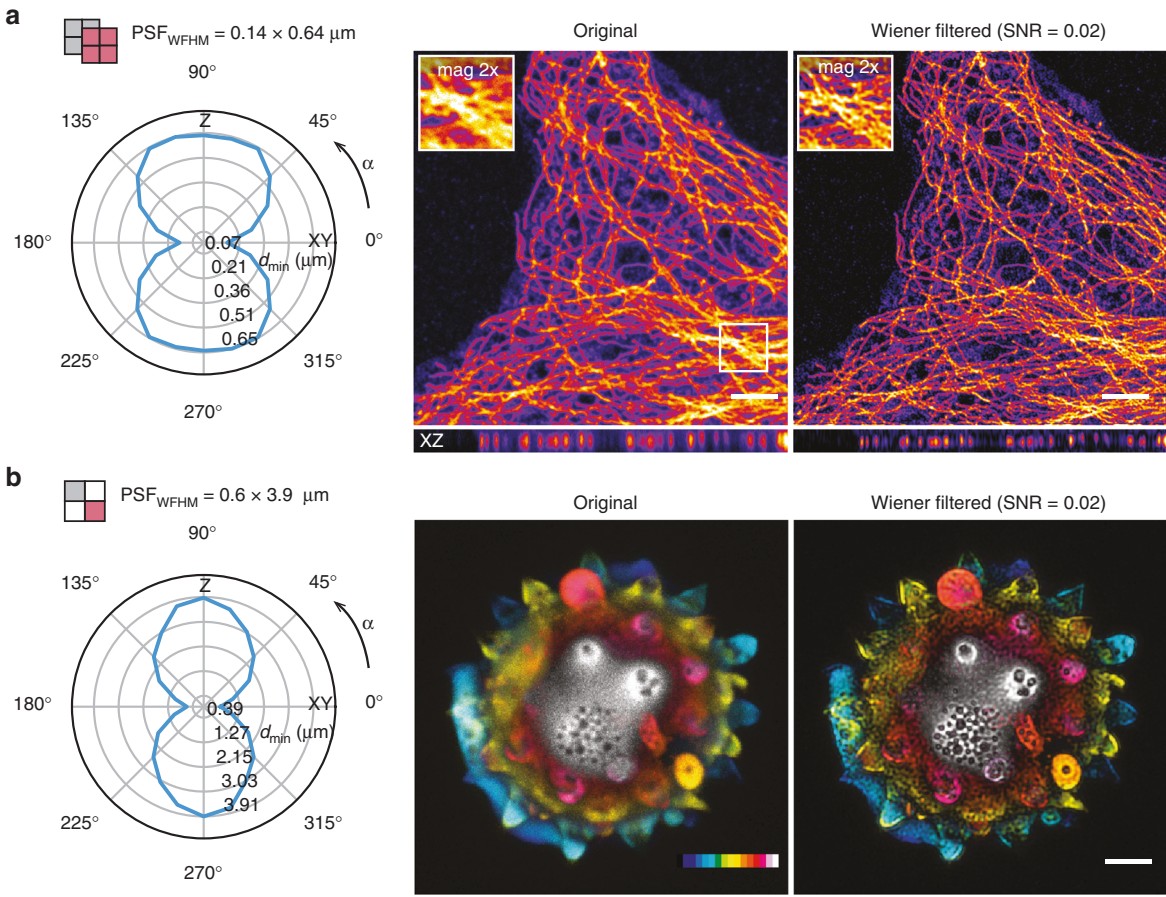

**Fig. 4** Blind Wiener deconvolution in 3D with PSF estimated from SFSC measurements. **a** Results of Wiener deconvolution of a super-resolution 3D image recorded with a STED microscope are shown. The results show dramatically improved effective resolution and reduction of axial haze. The SFSC estimate was calculated with two images. **b** Blind Wiener deconvolution result of a confocal image of Pollen are shown. The depth coded colormap reveals dramatic reduction of axial haze, and in general contrast and effective resolution are clearly improved. The polar plots illustrate the resolution point in micrometers, and it was calculated by rotating the SFSC section at 15° increments around the y-axis. Scale bars in **a** 3 μm, **b** 4 μm; depth color coding in **b** 20 μm

## Methods

**One and two-image resolution measurements with 2D images**. One- and two-image FRC measurements were both done according to

$$\text{FRC/FSC}_{12}(r_i) = \frac{\sum_{r \in r_i} F_1(r) \cdot F_2(r)^*}{\sqrt{\sum_{r \in r_i} F_1^2(r) \cdot \sum_{r \in r_i} F_2^2(r)}} \quad (2)$$

where $F_1$ and $F_2$ are the Fourier transforms of the two images and $r_i$ the $i$th frequency bin. Prior to the FRC calculations, in two-image case, the two images were registered by a phase-correlation based method[27]; in single-image case the splitting was performed to produce the four subimages. A Hamming window was applied to each image to suppress edge effects and other spurious correlations. With both one- and two-image FRC the 1/7 resolution threshold[8,9] was used to determine the numerical resolution value.

**Resolution measurements with 3D images**. Measurements on 3D images were done with our SFSC measure

$$\text{SFSC}_{12}(r_i, \alpha_i) = \frac{\sum_{r \in r_i, \alpha \in \alpha_i} F_1(r, \alpha, \phi) \cdot F_2(r, \alpha, \phi)}{\sqrt{\sum_{r \in r_i, \alpha \in \alpha_i} F_1^2(r, \alpha, \phi) \cdot \sum_{r \in r_i, \alpha \in \alpha_i} F_2^2(r, \alpha, \phi)}} \quad (3)$$

where $F_1(r, \alpha, \phi)$ and $F_2(r, \alpha, \phi)$ denote the voxels in two Fourier transformed images that are located (I) at a given distance $r_i$ from the origin and (II) within an orientation sector, defined by $\alpha$ and $\phi$ (Supplementary Fig. 9a). We compared the SFSC measures against FPC[9] as well as FRC measures.

The SFSC/FPC/FRC analyses with the 3D images were performed with the same logic as the FRC measures on the 2D images. Before data analysis each 3D image was resampled to isotropic spacing using linear interpolation. Single image splitting was achieved similarly to the 2D case; only one diagonal was used to limit the computational effort in SFSC/FPC. In the axial direction two consecutive (z) layers

were simply added together, to maintain the image proportions, and to not introduce additional offsets. Threshold curve based on $\text{SNR}_e = 0.5$ (one-bit) was used with SFSC, whereas with both FPC and FRC, 1/7 threshold produced more reasonable numerical values. The $\text{SNR}_e$ thresholds, proposed in[11] can be calculated from

$$T(r_i) = \frac{\text{SNR}_e + \left(2\sqrt{\text{SNR}_e} + 1\right)/\sqrt{N(r_i)}}{\text{SNR}_e + 1 + 2\sqrt{\text{SNR}_e}/\sqrt{N(r_i)}} \quad (4)$$

where $T(r_i)$ is the threshold value, $N(r_i)$ the number of pixels/voxels at $i$th Fourier ring/shell, and $\text{SNR}_e$ the expected SNR value at the cut off point.

Because of the anisotropic sampling that is typical to 3D fluorescence microscopy images, the frequency axes in SFSC and FRC curves need to be corrected to compensate for it. In this paper this was achieved by multiplying the image pixel/voxel size by factor $k(\theta)$:

$$k(\theta) = (1 + (z - 1)) \cdot |sin(\theta)| \quad (5)$$

where $z$ is the sampling anisotropy factor (e.g., two for an image sampled with half the sampling rate in depth ($Z$) with respect to the lateral ($XY$) direction) and $\theta$ denotes the rotation angle with respect to the $XY$ plane, which in case of SFSC is a multiple of $\alpha$. If no such correction is made, all the numerical resolution values calculated with FPC/FRC/SFSC at orientations $\theta \neq 0 + n\pi$ will be unrealistically high.

**Frequency domain low-pass filtering**. The frequency domain filtering was performed by first estimating the effective image resolution with FRC and then using it as a cut off frequency for a low-pass Fourier domain filter. Three different types of Fourier space filters were used in this work: (I) an ideal low-pass filter (ILPF), (II) a Butterworth low-pass filter, and (III) a Gaussian low-pass filter.

An ILPF can be defined as:

$$H(r_i) = \begin{cases} 1, & \text{if } r_i < r_{th} \\ 0, & \text{otherwise} \end{cases} \quad (6)$$

where $r_i$ is a polar distance from the center of the frequency domain filter (zero frequency) and $r_{th}$ is the distance at the cut off frequency, obtained with FRC. Frequencies after the cutoff are simply clipped to zero.

A Butterworth low-pass filter (BLPF) on the other hand is defined as:

$$H(r_i) = \frac{1}{1 + [r_i/r_{th}]^{2n}} \quad (7)$$

where $n$ denotes the degree of the filter, which controls how sharply the transition from pass-band (allowed frequencies) to stop-band (filtered frequencies) is made. When compared to ILPF, BLPF has nearly equally flat (unity) response in the pass-band, but transitions to stop-band more smoothly, thus avoiding a strong discontinuity at the cutoff; at the $r_{th}$, $H(r_i) = 0.5$.

A Gaussian low-pass filter (GLPF) is defined as:

$$H(r_i) = e^{-r_i^2/2r_{th}^2} \quad (8)$$

Similarly to BLPF, the GLPF has a smooth transition from pass-band to stop-band. The Gaussian function however, is not flat in the pass-band and transitions more slowly from pass-band to stop-band, which means that it may in some cases blur the filtered images and perform less than optimally, when filtering out noise. At the $r_{th}$, $H(r_i) = 0.607$.

**Image deconvolution**. Image formation in a microscope can be described as a convolution of every object sample point with the PSF:

$$i(x, y, z) = h(x, y, z) \otimes o(x, y, z) \quad (9)$$

where $i(x, y, z)$, $h(x, y, z)$, and $o(x, y, z)$ are the measured image, the PSF and the original sample object, respectively. By image deconvolution[28] one attempts to revert the blurring effect of the microscope, and thus increase image contrast and effective resolution, by using the PSF as a prior information. Deconvolution can be performed in a single step, e.g., by Wiener or Tikhonov filtering[29]—or then iteratively, e.g., by RL[30–32]. In this paper we use the Wiener and RL algorithms.

The Wiener deconvolution algorithm is based on inverse filtering, and takes advantage of the fact that a convolution operation in the spatial domain, becomes a multiplication in the frequency domain—and thus a relatively simple single-step deconvolution can be realized as follows:

$$O(u, v, w) = \left[ \frac{1}{H(u, v, w)} \frac{|H(u, v, w)|^2}{|H(u, v, w)|^2 + 1/\text{SNR}} \right] I(u, v, w) \quad (10)$$

where $O(u, v, w)$, $I(u, v, w)$, and $H(u, v, w)$ are the Fourier space representations of the estimate for the original object, the observed image and the PSF. $|H(u, v, w)|^2$ is the power spectrum of the PSF. The regularization term $\text{SNR}^{-1}$ can also be written as $|N(u, v, w)|^2/|O(u, v, w)|^2$, where $|N(u, v, w)|^2$ is the power spectrum of the noise and $|O(u, v, w)|^2$ is the power spectrum of the original object; neither of the two terms are known, which means that usually the value is decided on case-by-case basis, based on the subjective quality of the deconvolution results. The regularization factor is weighted by the power spectrum of the PSF. It will have a stronger effect at high frequencies, as the power spectrum approaches zero value.

The iterative RL algorithm can be described by

$$o_{k+1} = \left\{ \frac{i}{h \otimes o_k + b} \otimes h^* \right\} o_k \quad (11)$$

where $o_k$ and $o_{k+1}$ are the current and next object estimates, $i$ is the original image, $h$ is the PSF, $h^*$ its mirrored version (complex conjugate) and $b$ a background term. In the equation, the pixel indexes have been omitted to allow a simple presentation of the algorithm. $b$ is typically set to zero, but a nonzero value can be used to correct for a strong background signal. In our example (Supplementary Fig. 8) the background was estimated by first dividing an image into two segments (signal and background) with a simple intensity based spatial mask[33], after which the mean intensity value of the background segment was calculated and then used as $b$ in deconvolution.

With both Wiener filtering and RL, PSF estimate was generated on the basis of an FRC measurement on the original image data: the FRC resolution value was simply used as an FWHM value for a Gaussian PSF. With 3D images, separate values were used for lateral and axial directions. In the Adjustive RL algorithm the PSF is updated during RL iteration, by calculating the FRC resolution for each intermediate deconvolution estimate.

In addition to updating the PSF estimates (when so desired) the FRC measures of the intermediate estimates were used to observe the progress of the RL deconvolution. The deconvolution can be considered fully converged, when effective resolution reached its maximum value. However, as reaching the full convergence may take a large amount of iterations, we proposed two more practical thresholds based on the rate of change of the effective resolution ($-1$ nm it$^{-1}$ and $-0.2$ nm it$^{-1}$). We compared the FRC metric against two previously published ones $\tau_1$ and $\eta_k$[17–19,34].

With $\tau_1$ the relative difference between two subsequent deconvolution estimates is measured:

$$\tau_1 = \frac{\sum_v |o_k - o_{k-1}|}{\sum_v o_k} \quad (12)$$

where $o_k$ is the current estimate and $o_{k-1}$ the previous one. $\eta_k$ is a measure of convergence

$$\eta_k = \frac{N(u)}{N},$$
$$\left\{ u \in \frac{i}{h \otimes o_k} \otimes h^* \mid u < (0 - \varepsilon) \vee (0 + \varepsilon) < u < (1 - \varepsilon) \vee u > (1 + \varepsilon) \right\} \quad (13)$$

where $N$ is the total number of pixels, $N(u)$ is the number of pixels that are not currently converging and $\varepsilon$ is the convergence epsilon.

TV regularization[17] with fixed $\lambda_{TV} = 5 \times 10^{-4}$ was used as a smoothness constraint in our Adjustive RL-TV algorithm.

**Test images**. The test images consist of various types of confocal as well as STED microscope images that were acquired with a variety of commercial and custom-built microscopes. None of the samples was specifically prepared for this paper, but a short description for each is given in Supplementary Note 3.

## Data availability
The image data that supports the findings of this study are available in the *figshare* repository, https://doi.org/10.6084/m9.figshare.c.4511663.

## Code availability
The FRC/FSC measurement functions as well as all the image processing, analysis and data visualization tools used in this paper are available as open-source Python software library, called *Microscope Image Processing Library* (MIPLIB) at https://github.com/sakoho81/miplib. Please see Supplementary Note 4 for additional information about the software.

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

## Acknowledgements

The authors thank Dr. Paolo Bianchini and Prof. Colin J. R. Sheppard (Istituto Italiano di Tecnologia) for useful discussion, and Elena Tcarenkova (University of Turku) for help with obtaining the Abberior STED images.

## Author contributions

S.K., G.T., M.C., and G.V. conceived the idea. S.K. and G.V. planned the studies and the experiments. S.K. wrote the software and performed the majority of the experiments. T. D. performed the three-dimensional confocal experiments. S.K. and G.V. analyzed the data. S.K., G.T., M.C., T.D., A.D. and G.V. participated in extensive discussions during the course of the research. S.K. and G.V. wrote the manuscript. G.T., M.C., T.D., and A. D. participated in revising the manuscript.

## Additional information

**Competing interests:** The authors declare no competing interests.

**Peer Review Information:** *Nature Communications* thanks Niccolo Banterle, Ryan McGorty and other anonymous reviewer(s) for their contribution to the peer review of this work. Peer reviewer reports are available.

