## [Peer Review File · Nature Communications]

Reviewers' comments:

Reviewer #1 (Remarks to the Author):

Koho et al report on the use of the FRC image resolution measure for image filtering and deconvolution purposes, and introduce two new technicalities in order to enable this, first of all a method of splitting a single image in two noise independent sub-images, second a new way to deal with the anisotropic resolution in 3D microscopy images. The ideas are nice and useful, and can find broader application in microscopy image analysis. I can, however, not recommend publication of the manuscript. I have one major issue and several minor issues:

1. The image splitting procedure is not correctly tested/evaluated, and is in my opinion basically flawed or has limited applicability. The key idea is to split every 2x2 set of pixels in two parts, taking the +45 deg diagonal as sub-image 1 and the -45 deg diagonal as sub-image 2. The missing pixels are then filled in via interpolation. In any case, this is simply sub-sampling and inevitably induces loss of information, unless the original image is significantly oversampled. This latter condition is a necessary condition for the authors' method to work, but is not stated anywhere, nor is the effect of undersampling/oversampling methodically assessed. This is an absolute requirement for the authors to address. Reference [8] already points to the impact of (under)sampling on FRC. Basically, pixilation/sampling at a length scale b induces a damping of signal content as a function of spatial frequency as $\text{sinc}(\pi \cdot q \cdot b)^2$, with q the spatial frequency. A rule-of-thumb in reference [8] of having $b \sim 1/4/q_0$, with q_0 the spatial frequency at the FRC threshold gives a damping of about 80% and seems tolerable. For the STED-example in fig. 2a this means that a sampling distance of about 20 nm would be OK, but somewhat bigger would not. In fig. 2b the difference between the multi-image and single-image FRC is already a bit bigger, which may be just this effect going on, as the FRC resolutions are smaller than in fig. 2a but the sampling distance is probably not.

2. Crucial information on the data is missing. In particular, because of my first remark, I am missing data on pixel sizes/sampling distances.

3. The structure of the paper is often unclear and the figures also lack clarity.

- For example, the explanation of the image splitting in fig. 1 is not very insightful. Actually I am guessing that the missing pixels of the two diagonal sub-lattices are filled in by interpolation (if so, how?).

- An additional rescaling operation with $\sqrt{2}$ is also not well explained. Perhaps adding some explicit math in the supplement can help, it is a bit too much hand-waving to my liking now.

- The same holds for sfigure 1, which took me some time to comprehend.

- Another point is the introduction of S-FSC based on the wedge-based division of Fourier space. Now, nothing about this is explained in the main text, I suggest moving some qualitative explanations from the supplement to the main text and adding more quantitative, mathematically formulated information in the supplement.

- The meaning of the second and third column in fig. 6 only becomes clear after digging into the supplement. It would be way more useful to see only the left curves in fig. 6a and 6b and some representative images before, at, and beyond the optimum of the iteration.

4. The SNR-parameter in fig. 4 is ad-hoc I take it. Would it vary from image to image or is this a uniquely set parameter?

5. The assessment of the 3D-variant of FRC introduced in ref. [8] is not correct. It does not rely on subjectively, manually selecting slices for FRC evaluation as claimed. In fact, what it does is to average over all possible planes in Fourier space (arbitrary direction and distance from the origin). To that end it requires a lot of image rotations (for each direction of the plane) and a 3D Fourier transform per rotation. This makes it numerically complex and time consuming, which is a significant drawback. It does not need any human intervention.

6. Ref. [20] for theoretically motivated PSFs is not appropriate. I suggest the authors look into the considerable literature on physically motivated PSF models, based on vectorial imaging of dipole emitters.

7. In Eq. (9) there is an overall division by $H(u,v,w)$, the transfer function of the microscope. I think it should not be there.

Reviewer #2 (Remarks to the Author):

In the manuscript "Fourier ring/Shell correlation simplifies image restoration in fluorescence microscopy" the authors present a novel application of Fourier Ring Correlation as a mean to optimally filter/deconvolve microscopy images. The first and key result of the manuscript is the finding that FRC is suitable not only when two independent instance of an image are available (such as the two independent reconstruction of a single molecule based super-resolution data-set) but also in the case of single images by a pixel-wise splitting for which they propose a strategy. This finding allows to expand the application domain of FRC, currently primarily used to assess the resolution of microscopy images. Specifically the author explore filtering alternative to gaussian (namely Ideal and Butterworth) as well as takes the concept to the further step of using FRC to provide a good estimate of the PSF used in the deconvolution process (testing both blind Wiener and Richardson-Lucy deconvolution) and provide a stopping criterium for the iterative deconvolution processes. Finally the filtering of anisotropic three dimensional dataset is addressed. The extension of FRC to single images is an important finding and the combination with optimal filtering/deconvolution and extension to 3D datasets could find a widespread use in the imaging community and is worthy of publication. In order to consider publication is however fundamental to prove the generality of the method. In this regard I have two main separate point which should be addressed regarding two separate aspect of the manuscript; specifically the single image FRC and the application of FRC to filtering/deconvolution.

1) The possibility to apply FRC to single images is convincingly demonstrated in the paper and the strategy proposed clearly outlined (I appreciated that the author provide a python library). What is less clearly addressed are the theoretical and practical limitation of the proposed strategy. Specifically the authors show that when the splitting is performed along the fast scanning axis single image FRC becomes artifactual due to the residual correlation between neighboring pixels. This suggests that the applicability of the technique is bounded by intra-pixel correlation which in the presented images are null along the slow scan axis. Would this requirement always hold? How slow does the slow scan axis needs to be to assure pixel along it are uncorrelated? At the limit case is the same strategy applicable also to non-scanning imaging technique? If so the author should provide an example, if not it should be clearly stated. In general placing the single image FRC in the context of image formation theory would significantly improve the manuscript by defining its natural boundary of applicability.

2) The authors clearly show that single image FRC is powerful tool to achieve optimal deconvolution. A very promising consequence investigated in the manuscript is that FRC can also be used to determine the number of iterations in the deconvolution process. In this case the clear risk is that artifactual frequency introduced by the deconvolution process might correlate possibly leading to overestimation in the optimal number of iterations. Indeed this effect is observed and discussed in the case of two image FRC. While in the particular image(s) analyzed this is not the case for single image FRC, the argument of the authors that this is because "the two sub-images contain different pixels, and thus noise generated by deconvolution, does not easily co-localize" albeit likely correct is more hand-waving than an actual proof. Given the importance of this point for the generality of the method the author should either provide a proof that is always the case or alternatively provide a mean to automatically identify artifactual resolutions by e.g. a systematic comparison of the FRC curve changes in following iterations (or by quantifying at each step the "goodness" of the obtained FRC curve).

Minor points:

- While the supplementary material is rather exhaustive Supplementary Fig. 9 is rather poorly labeled and explained. I understood that the sample is the same just imaged at different STED power and detectors, however the image shown in the upper right panel (Detector 2, 100% STED)

shows a very marked different feature on the up-right corner in the image. Why is that visible only with detector 2?

- I personally find confusing the change in the color scale in Fig. 4 compared to all other figures
- The title of Supplementary Fig. 3 has a typo
- Minor typo in the Abbreviation list ("FRC," should be "FRC:")
- In Supplementary Table 1 method d) the meaning of the no/no in the last column is not clear.

Reviewer #3 (Remarks to the Author):

In this paper, the authors show that Fourier ring correlation (FRC) / Fourier shell correlation (FSC) can be used for multiple different tasks. While FRC/FSC have been previously used to quantify image resolution in different microscopy techniques, the authors here describe other ways FRC/FSC can be used. In particular, they use FRC/FSC to filter or deconvolve images, resulting in higher-quality images. Some of the images the authors show are impressive. I believe the uses of FRC/FSC the authors describe would be of value to many other researchers. However, I had a difficult time following the finer points of the paper and fully understanding what was done. I found the organization of the paper confusing.

I'll start with two general suggestions for improving the paper.

1. Organization.

a. I found the flow of the paper did not help in understanding what was going on. The 3rd section under "Results" (Blind Wiener Deconvolution in 2D/3D) references material in the following section (Blind Iterative Richardson Lucy Deconvolution with FRC based PSF). I had to read over these sections multiple times to grasp what was being done.

b. For some figures, I think restructuring would improve the presentation. I did not find Figure 1 to be very informative (certainly not given its large footprint). In some figures (like Fig 3), it would be nice if the frequency range on the x-axis were the same in the multiple plots of correlation vs. frequency.

2. Motivation.

a. The authors state that for FRC one needs to acquire two images of the sample and that this can be problematic. However, I'm not sure that this is really much of a problem. For most biological imaging where the SNR may be low (and so where deconvolution and the methods described in this paper would be most useful) would be for live cell imaging. In such cases, it is likely that there will be time lapses recorded and so multiple images will be available. Said another way, in cases where the exposure/acquisition time must be kept to a minimum tend to be cases where higher exposure times would cause cell death or photobleaching. Cases where cell death or photobleaching are of a concern, tend to be cases where movies are acquired, not single images. However, there may be situations where the sample is very dynamic and a movie needs to be acquired. In such cases, there may be substantial frame-to-frame differences. I can see that as a situation where FRC would be needed to be done on a single frame and another frame showing the sample exactly as it was would be impossible to acquire.

The authors may want to read: "High-resolution restoration of 3D structures from widefield images with extreme low signal-to-noise ratio," Muthuvel Arigovindan, Jennifer C. Fung, Daniel Elnatan, Vito Mennella, Yee-Hung Mark Chan, Michael Pollard, Eric Brärlund, John W. Sedat, and David A. Agard, PNAS, 2013, Oct 22, 110(43), 17344-9.

In that article, Arigovindan et al. indeed use FRC to compare different deconvolution algorithms. The authors here could also look at the deconvolution results in that paper which are generally more impressive than the results presented here. Perhaps if the authors here acquire images at lower SNR and show the results of their deconvolution using FRC to get a PSF, then there could be equally impressive results. However, I recognize that that may require taking new data and that might not be feasible.

Other, specific comments on the article:

3. The authors claim that the PSF estimated from FRC is “actually rather close to the optimal, as otherwise either blurring or strong increase in noise should be expected.” ♦ Could the authors use another method of finding size of the PSF, for example by measuring line profile across a microtubule and deconvolving that with the known width of microtubules? Alternatively, the authors could acquire a z-stack using sub-resolution-sized beads to acquire a PSF. There are multiple ways the authors could back up their claim that FRC is giving them an accurate PSF measurement.

a. If possible, I wonder if FRC could be used to see how the PSF varies with defocus. Could the authors acquire slightly out-of-focus images and, from FRC, see how the width of the PSF varies with defocus. Perhaps see how that PSF width varies with positive or negative defocus as some amount of spherical aberration is likely present. That could be a compelling result.

4. Explain rationale for using one-bit as a resolution threshold (for eqn 2). I didn't quite follow the explanation given at around line 276.

5. Nieuwenhuizen et al. (whom the authors here cite) found that, using Fourier correlation, the measured resolution was different in directions parallel to versus orthogonal to cellular filaments. In this paper, Koho et al. write that an advantage of FRC/FSC is that it gives “image-independent” results (line 22). Could the authors explain what they mean by “image-independent” and whether that contradicts what others have found regarding a dependence of FRC on the structures imaged?

6. More on this point of different results for FRC parallel to or orthogonal to imaged filaments. The image used in Figures 3 and 4 is of a microtubulin stained cell. The filaments seems to mostly run at an approx. 45 degree angle from top right to bottom left. The way the image is split to do the FRC uses pixels along an orthogonal line (the pixel in the upper left and the pixel in the lower right of the 2x2 group are used). Could the authors try using the lower left and upper right pixels instead?

7. Continuing on this point of trying different image splitting schemes. The authors try diagonal splitting ((a) in Supplemental Note 5). They also try diagonal summing ((d) in Supplemental Note 5). It seems to me the authors might want to try doing the diagonal splitting (a), but do so twice (once in each diagonal direction). They could then average the FRC results if the two directions are similar. Or if the results from the different diagonal splittings are different, the authors should explain why that could be.

8. In lines 347-349, the authors state that a blind deconvolution algorithm can be used in which case both the object and the PSF are estimated iteratively. Is that what the authors here are doing? That is, are the authors using FRC to first estimate the PSF and then using a blind deconvolution algorithm to improve on the PSF estimate as well as on the imaged object? From Supplemental Note 3, it seems that the authors are not updating the estimate of the PSF in their deconvolution.

9. What are the sizes of the PSF used in the deconvolutions. In Supplementary Figure 5, deconvolution is done with FRC based PSF and a theoretical PSF. What are the widths of those? It would be good to know. It would also help to know how sensitive the deconvolution the authors are using is to the width of the given PSF. Would the results change much if a different FRC threshold were used (such as 1/7 instead of the current threshold)?

10. In regards to dealing with three-dimensional images with anisotropic resolution.

Nieuwenhuizen et al. (2013) use Fourier line correlation (FLC) or Fourier plane correlation (FPC) to deal with anisotropic image resolution. How do those methods compare to the sectioned FSC the authors employ?

11. What was the image size used for FRC/SFRC analysis (in pixels)? Could the authors use their methods to find the resolution in different areas of an image? For instance, they could use their methods on 128x128 regions and see how the resolution varies.

12. PSF estimation. Would the authors be able to measure PSF from FRC on different subregions of the image? This might back up the claims made in the section starting on line 325. Could they show that the PSF is different in regions where the cell is thicker in comparison to where the cell is thinner (supporting idea that cell introduces aberrations). See, for example, Supplemental Figure 12 in Nieuwenhuizen et al. (2013).

13. In Figure 6, it would be nice if the tau and eta parameters were defined. Units should be given in the resolution plots (units for resolution missing in other plots too).
14. Supplemental Figure 9. The different labels "Det1_STED_###" mean nothing to the reader. Could the authors explain what these different images are.
15. Supplemental Figure 9. Could the authors use their deconvolution methods (with FRC-based PSF) to improve the resolution of, for example, Det2_STED_50? If they could run FRC analysis between the highest resolution image (like Det1_STED_100) and a lower-resolution image before and after the lower-resolution image is deconvolved using FRC-based PSF (similar in spirit to what Arigovindan 2013 did), that would be nice.

Overall, interesting results! I just think it was hard to follow and some details not fully transparent (at least to me but maybe they are obvious to others).

In this paper, the authors show that Fourier ring correlation (FRC) / Fourier shell correlation (FSC) can be used for multiple different tasks. While FRC/FSC have been previously used to quantify image resolution in different microscopy techniques, the authors here describe other ways FRC/FSC can be used. In particular, they use FRC/FSC to filter or deconvolve images, resulting in higher-quality images. Some of the images the authors show are impressive. I believe the uses of FRC/FSC the authors describe would be of value to many other researchers. However, I had a difficult time following the finer points of the paper and fully understanding what was done. I found the organization of the paper confusing.

I'll start with **two general suggestions** for improving the paper.

1. Organization.

- a. I found the flow of the paper did not help in understanding what was going on. The 3rd section under "Results" (Blind Wiener Deconvolution in 2D/3D) references material in the following section (Blind Iterative Richardson Lucy Deconvolution with FRC based PSF). I had to read over these sections multiple times to grasp what was being done.
- b. For some figures, I think restructuring would improve the presentation. I did not find Figure 1 to be very informative (certainly not given its large footprint). In some figures (like Fig 3), it would be nice if the frequency range on the x-axis were the same in the multiple plots of correlation vs. frequency.

2. Motivation.

- a. The authors state that for FRC one needs to acquire two images of the sample and that this can be problematic. However, I'm not sure that this is really much of a problem. For most biological imaging where the SNR may be low (and so where deconvolution and the methods described in this paper would be most useful) would be for live cell imaging. In such cases, it is likely that there will be time lapses recorded and so multiple images will be available. Said another way, in cases where the exposure/acquisition time must be kept to a minimum tend to be cases where higher exposure times would cause cell death or photobleaching. Cases where cell death or photobleaching are of a concern, tend to be cases where movies are acquired, not single images. However, there may be situations where the sample is very dynamic and a movie needs to be acquired. In such cases, there may be substantial frame-to-frame differences. I can see that as a situation where FRC would be needed to be done on a single frame and another frame showing the sample exactly as it was would be impossible to acquire.

The authors may want to read: "High-resolution restoration of 3D structures from widefield images with extreme low signal-to-noise ratio," Muthuvel Arigovindan, Jennifer C. Fung, Daniel Elnatan, Vito Mennella, Yee-Hung Mark Chan, Michael Pollard, Eric Branlund, John W. Sedat, and David A. Agard, *PNAS*, 2013, Oct 22, 110(43), 17344-9.

In that article, Arigovindan et al. indeed use FRC to compare different deconvolution algorithms. The authors here could also look at the deconvolution results in that paper which are generally more impressive than the results presented here. Perhaps if the authors here acquire images at lower SNR and show the results of their deconvolution using FRC to get a PSF, then there could be equally impressive results. However, I recognize that that may require taking new data and that might not be feasible.

Other, specific comments on the article:

3. The authors claim that the PSF estimated from FRC is “actually rather close to the optimal, as otherwise either blurring or strong increase in noise should be expected.” → Could the authors use another method of finding size of the PSF, for example by measuring line profile across a microtubule and deconvolving that with the known width of microtubules? Alternatively, the authors could acquire a z-stack using sub-resolution-sized beads to acquire a PSF. There are multiple ways the authors could back up their claim that FRC is giving them an accurate PSF measurement.
 - a. If possible, I wonder if FRC could be used to see how the PSF varies with defocus. Could the authors acquire slightly out-of-focus images and, from FRC, see how the width of the PSF varies with defocus. Perhaps see how that PSF width varies with positive or negative defocus as some amount of spherical aberration is likely present. That could be a compelling result.
4. Explain rationale for using one-bit as a resolution threshold (for eqn 2). I didn't quite follow the explanation given at around line 276.
5. Nieuwenhuizen et al. (whom the authors here cite) found that, using Fourier correlation, the measured resolution was different in directions parallel to versus orthogonal to cellular filaments. In this paper, Koho et al. write that an advantage of FRC/FSC is that it gives “image-independent” results (line 22). Could the authors explain what they mean by “image-independent” and whether that contradicts what others have found regarding a dependence of FRC on the structures imaged?
6. More on this point of different results for FRC parallel to or orthogonal to imaged filaments. The image used in Figures 3 and 4 is of a microtubulin stained cell. The filaments seems to mostly run at an approx. 45 degree angle from top right to bottom left. The way the image is split to do the FRC uses pixels along an orthogonal line (the pixel in the upper left and the pixel in the lower right of the 2x2 group are used). Could the authors try using the lower left and upper right pixels instead?
7. Continuing on this point of trying different image splitting schemes. The authors try diagonal splitting ((a) in Supplemental Note 5). They also try diagonal summing ((d) in Supplemental Note 5). It seems to me the authors might want to try doing the diagonal splitting (a), but do so twice (once in each diagonal direction). They could then average the FRC results if the two directions are similar. Or if the results from the different diagonal splittings are different, the authors should explain why that could be.
8. In lines 347-349, the authors state that a blind deconvolution algorithm can be used in which case **both** the object and the PSF are estimated iteratively. Is that what the authors here are doing? That is, are the authors using FRC to first estimate the PSF and then using a blind deconvolution algorithm to improve on the PSF estimate as well as on the imaged object? From Supplemental Note 3, it seems that the authors are **not** updating the estimate of the PSF in their deconvolution.
9. What are the sizes of the PSF used in the deconvolutions. In Supplementary Figure 5, deconvolution is done with FRC based PSF and a theoretical PSF. What are the widths of those? It would be good to know. It would also help to know how sensitive the deconvolution the

authors are using is to the width of the given PSF. Would the results change much if a different FRC threshold were used (such as 1/7 instead of the current threshold)?

10. In regards to dealing with three-dimensional images with anisotropic resolution. Nieuwenhuizen et al. (2013) use Fourier line correlation (FLC) or Fourier plane correlation (FPC) to deal with anisotropic image resolution. How do those methods compare to the sectioned FSC the authors employ?
11. What was the image size used for FRC/SFRC analysis (in pixels)? Could the authors use their methods to find the resolution in different areas of an image? For instance, they could use their methods on 128x128 regions and see how the resolution varies.
12. PSF estimation. Would the authors be able to measure PSF from FRC on different subregions of the image? This might back up the claims made in the section starting on line 325. Could they show that the PSF is different in regions where the cell is thicker in comparison to where the cell is thinner (supporting idea that cell introduces aberrations). See, for example, Supplemental Figure 12 in Nieuwenhuizen et al. (2013).
13. In Figure 6, it would be nice if the tau and eta parameters were defined. Units should be given in the resolution plots (units for resolution missing in other plots too).
14. Supplemental Figure 9. The different labels "Det1_STED_##" mean nothing to the reader. Could the authors explain what these different images are.
15. Supplemental Figure 9. Could the authors use their deconvolution methods (with FRC-based PSF) to improve the resolution of, for example, Det2_STED_50? If they could run FRC analysis between the highest resolution image (like Det1_STED_100) and a lower-resolution image before and after the lower-resolution image is deconvolved using FRC-based PSF (similar in spirit to what Arigovindan 2013 did), that would be nice.

Answers to Reviewers

We thank the Referees for their interest in our work and for great comments that helped us improve our manuscript. We have checked all the comments provided by the Referees and have made necessary changes accordingly to their indications.

Reviewer #1:

Comment 1:

The image splitting procedure is not correctly tested/evaluated, and is in my opinion basically flawed or has limited applicability. The key idea is to split every 2×2 set of pixels in two parts, taking the $+45$ deg diagonal as sub-image 1 and the -45 deg diagonal as sub-image 2. The missing pixels are then filled in via interpolation. In any case, this is simply sub-sampling and inevitably induces loss of information, unless the original image is significantly oversampled. This latter condition is a necessary condition for the authors' method to work, but is not stated anywhere, nor is the effect of undersampling/oversampling methodically assessed. This is an absolute requirement for the authors to address. Reference [8] already points to the impact of (under)sampling on FRC. Basically, pixilation/sampling at a length scale b induces a damping of signal content as a function of spatial frequency as $\text{sinc}(\pi q b)^2$, with q the spatial frequency. A rule-of-thumb in reference [8] of having $b \sim 1/4/q_0$, with q_0 the spatial frequency at the FRC threshold gives a damping of about 80% and seems tolerable. For the STED-example in fig. 2a this means that a sampling distance of about 20 nm would be OK, but somewhat bigger would not. In fig. 2b the difference between the multi-image and single-image FRC is already a bit bigger, which may be just this effect going on, as the FRC resolutions are smaller than in fig. 2a but the sampling distance is probably not.

Answer:

Thank you for an excellent and very detailed comment that helped us to significantly improve the manuscript. We revised the whole text with these comments in mind, and most importantly added a completely new section to the beginning of the **Results** section that focuses on the characterization of the single image splitting method.

As our results show, due to the peculiar modulation effect that the diagonal single image splitting introduces, it is actually possible to use the method up-to the Nyquist limit (we also added a mathematical description for this effect). In fact the splitting method as it is implemented now, has been tuned to work best at "normal" sampling distances $d_{min}/3 - d_{min}/5$. At higher frequencies, as pointed out by the referee, and as our results show as well, the FRC curve indeed dampens. It is possible to compensate for this, if one so desires, by using a lower threshold level (e.g. the $1/7$ as suggested in the manuscript), or then one just keeps in mind, that there is a slight underestimation in the resolution values near to the sampling limit. On the other hand, with significantly oversampled data, the resolution appears to be slightly overestimated at the $\text{SNR}_e = 0.25$, but once again the differences are rather minor, and as shown in a RL deconvolution example with an image with $d_{min}/8$ pixel size, this does not seem to be wrong or an issue in practical applications.

One could of course think of implementing a specific threshold curve for the one image splitting which would slope from ≈ 0.4 at $1/4$ to ≈ 0.15 at $3/4$ of the frequency range, in order to match the numerical values with the $1/7$ limit in the two-image FRC.

Comment 2:

Crucial information on the data is missing. In particular, because of my first remark, I am missing data on pixel sizes/sampling distances.

Answer:

We added the pixel size/image dimension information for each image shown in the manuscript

Comment 3:

The structure of the paper is often unclear and the figures also lack clarity.

- For example, the explanation of the image splitting in fig. 1 is not very insightful. Actually I am guessing that the missing pixels of the two diagonal sub-lattices are filled in by interpolation (if so, how?).
- An additional rescaling operation with $\sqrt{2}$ is also not well explained. Perhaps adding some explicit math in the supplement can help, it is a bit too much hand-waving to my liking now.
- The same holds for sfigure 1, which took me some time to comprehend.
- Another point is the introduction of S-FSC based on the wedge-based division of Fourier space. Now, nothing about this is explained in the main text, I suggest moving some qualitative explanations from the supplement to the main text and adding more quantitative, mathematically formulated information in the supplement.
- The meaning of the second and third column in fig. 6 only becomes clear after digging into the supplement. It would be way more useful to see only the left curves in fig. 6a and 6b and some representative images before, at, and beyond the optimum of the iteration.

Answer:

We have revised the whole manuscript with these comments in mind, and structured it in a way to be more suitable for Nature format publication. Many of the important descriptive details were indeed somewhat hidden inside the Methods and Supplementary Materials.

The single image splitting method, as already discussed in *Comment 1* is now more accurately described. We also in detail discuss the motivations for the $\sqrt{2}$ correction, and show by examples that it indeed makes it possible to use the one-image FRC up-to rather sparse sampling distances.

The Deconvolution section was also completely revised. The former *Figure 6* was revised according to the reviewers suggestions, and in the end we moved it entirely to Supplementary Material (*Figure S. 9*). We also added several other deconvolution examples, as well as a worst case simulation of sorts to the Supplementary Material.

The description for the wedge based SFSC was now added to the main text.

Comment 4:

The SNR-parameter in fig. 4 is ad-hoc I take it. Would it vary from image to image or is this a uniquely set parameter?

Answer:

Yes, the *SNR* parameter that we use to regularize the linear Wiener filtering was selected on case-by-case basis, based on subjective quality of the deconvolution results. In practice, usually a value 0.1-0.01 works just fine with 2D images, and we seldom adjust it.

It might of course be possible to try to estimate the regularization factor from the data, e.g. by using simple intensity based spatial masking (Koho et al. 2016) to separate the areas with details (signal) and the background and comparing the average brightness of those areas. We did not try that, but it might be worth doing to hide one more parameter from the user.

Comment 5:

The assessment of the 3D-variant of FRC introduced in ref. [8] is not correct. It does not rely on on subjectively, manually selecting slices for FRC evaluation as claimed. In fact, what it does is to average over all possible planes in Fourier space (arbitrary direction and distance from the origin). To that end it requires a lot of image rotations (for each direction of the plane) and a 3D Fourier transform per rotation. This makes it numerically complex and time consuming, which is a significant drawback. It does not need any human intervention.

Answer:

Indeed we had not read the description properly. Thank you for pointing that out. We implemented the FPC algorithm in MIPLIB and included comparisons to that as well.

Comment 6:

Ref. [20] for theoretically motivated PSFs is not appropriate. I suggest the authors look into the considerable literature on physically motivated PSF models, based on vectorial imaging of dipole emitters.

Answer:

Yes, indeed the Gaussian approximations in (Zhang et al, 2007) are rather simplistic and we are aware of the rich literature from Richards & Wolf onwards that is available, regarding optical systems theory and theoretical PSFs. Our aim was not to devalue the importance of these works in any way, but rather to point out that obtaining a valid theoretical PSF requires a lot of prior knowledge. We now removed the whole discussion about the theoretical PSFs, in order to not get too far side tracked from the work presented here.

Comment 7:

In Eq. (9) there is an overall division by $H(u,v,w)$, the transfer function of the microscope. I think it should not be there.

Answer:

In the Eq. (11) (formerly 9) the division $O(u, v, w) = I(u, v, w)/H(u, v, w)$ is the inverse filter. The right side of the equation is simply the regularization term, as in a normal Wiener filter. The equation was written according to (Gonzalez & Woods, Digital Image Processing, 3rd edition).

Reviewer #2:

Comment 8:

The possibility to apply FRC to single images is convincingly demonstrated in the paper and the strategy proposed clearly outlined (I appreciated that the author provide a python library). What is less clearly addressed are the theoretical and practical limitation of the proposed strategy. Specifically the authors show that when the splitting is performed along the fast scanning axis single image FRC becomes artifactual due to the residual correlation between neighboring pixels. This suggests that the applicability of the technique is bounded by intra-pixel correlation which in the presented images are null along the slow scan axis. Would this requirement always hold? How slow does the slow scan axis needs to be to assure pixel along it are uncorrelated? At the limit case is the same strategy applicable also to non-scanning imaging technique? If so the author should provide an example, if not it should be clearly stated. In general placing the single image FRC in the context of image formation theory would significantly improve the manuscript by defining its natural boundary of applicability.

Answer:

Please refer to *Comment 1*, as we already discussed most of the aspects there. We now focus rather exclusively to the diagonal splitting pattern, as it is the only one that works with normal sampling densities and with all kinds of images. The diagonal splitting does not appear to be at all sensitive to slow/fast axis residual correlations (more than a regular FRC would be) -- we tried with rather significantly oversampled data ($d_{min}/8 - d_{min}/9$). The method also works in widefield -- we added an example of that.

Comment 9:

The authors clearly show that single image FRC is powerful tool to achieve optimal deconvolution. A very promising consequence investigated in the manuscript is that FRC can also be used to determine the number of iterations in the deconvolution process. In this case the clear risk is that artifactual frequency introduced by the deconvolution process might correlate possibly leading to overestimation in the optimal number of iterations. Indeed this effect is observed and discussed in the case of two image FRC. While in the particular image(s) analyzed this is not the case for single image FRC, the argument of the authors that this is because “the two sub-images contain different pixels, and thus noise generated by deconvolution, does not easily co-localize” albeit likely correct is more hand-waving than an actual proof. Given the importance of this point for the generality of the method the author should either provide a proof that is always the case or alternatively provide a mean to automatically identify artifactual resolutions by e.g. a systematic comparison of the FRC curve changes in following iterations (or by quantifying at each step the “goodness” of the obtained FRC curve).

Answer:

Thank you for an excellent comment. This helped us to improve the part about observing deconvolution progress significantly. We also added a worst-case simulation of sorts in the Supplementary Material, in which we show that it is indeed possible to see the noise bump in the single image splitting as well. It did not in the simulation prevent the $resolution = f(iteration)$ observer from working correctly, but in any case we introduced a second derivative of the said curve as an alternative stopping condition, to detect abnormal convergence behavior. We have not

managed to recreate a similar situation with a cell sample – it would probably be possible with a combination of significantly oversampled image (to have the necessary bandwidth for the noise bump), with a strong constant background and a significantly too small PSF. But then again, everything has its limits. We did test deconvolution with oversampled data $d_{min}/8$ (Figure S. 6), but did not see a trace of that.

Comment 10:

While the supplementary material is rather exhaustive Supplementary Fig. 9 is rather poorly labeled and explained. I understood that the sample is the same just imaged at different STED power and detectors, however the image shown in the upper right panel (Detector 2, 100% STED) shows a very marked different feature on the up-right corner in the image. Why is that visible only with detector 2?

Answer:

We were intentionally a little bit vague with the description, as we did not want to in any way claim that one detector configuration in the Leica microscope would be superior to the other. Also the data is not good enough in any case to make such conclusions. The two detectors APD and HyD in the Leica system are on different detection paths, and different filters are used. The (Detector 2, 100%) image appears to show some kind of a reflection effect that sometimes becomes visible at high depletion laser powers -- it has to do with the detection filter / depletion wavelength configuration. We replaced the said image with (Detector 2, 90% STED) to avoid confusion. The idea of the figure is to outline that the image resolution (FRC) parameter can be combined with other image quality related measures, such as contrast (Entropy), noisiness (fSTD) etc, to produce quantitative measures of image quality, to answer: Which one is the best image?

Comment 11:

I personally find confusing the change in the color scale in Fig. 4 compared to all other figures

Answer:

We changed all the LUTs to be the same with the HeLa cell tubulin image in the main text now.

Comment 12

- The title of Supplementary Fig. 3 has a typo
- Minor typo in the Abbreviation list (“FRC,” should be “FRC:”)

Answer:

After all the changes, we hope that we have not introduced too many new ones; these were fixed at least.

Comment 13

- In Supplementary Table 1 method d) the meaning of the no/no in the last column is not clear.

It means that the splitting method f) is

- asymmetrically sampled, because each new pixel consists of two rows and one column
- sensitive to the fast axis fluctuations, as the image splitting is done in the direction of the fast axis

Reviewer #3 :

Comment 14:

Organization. a. I found the flow of the paper did not help in understanding what was going on. The 3rd section under “Results” (Blind Wiener Deconvolution in 2D/3D) references material in the following section (Blind Iterative Richardson Lucy Deconvolution with FRC based PSF). I had to read over these sections multiple times to grasp what was being done. b. For some figures, I think restructuring would improve the presentation. I did not find Figure 1 to be very informative (certainly not given its large footprint). In some figures (like Fig 3), it would be nice if the frequency range on the x-axis were the same in the multiple plots of correlation vs. frequency.

Answer:

Thank you for an excellent comment. The structure of the article indeed was not very suitable, as many of the important details were hidden in the Methods as well as Supplementary Materials. We have re-written the whole manuscript, and hope that it is much more understandable now. We also revised all the figures. Most of the FRC curves are also displayed on a common frequency axis now.

Comment 15:

Motivation. a. The authors state that for FRC one needs to acquire two images of the sample and that this can be problematic. However, I'm not sure that this is really much of a problem. For most biological imaging where the SNR may be low (and so where deconvolution and the methods described in this paper would be most useful) would be for live cell imaging. In such cases, it is likely that there will be time lapses recorded and so multiple images will be available. Said another way, in cases where the exposure/acquisition time must be kept to a minimum tend to be cases where higher exposure times would cause cell death or photobleaching. Cases where cell death or photobleaching are of a concern, tend to be cases where movies are acquired, not single images. However, there may be situations where the sample is very dynamic and a movie needs to be acquired. In such cases, there may be substantial frame-to-frame differences. I can see that as a situation where FRC would be needed to be done on a single frame and another frame showing the sample exactly as it was would be impossible to acquire.

Answer:

Indeed, two images are often available, especially in widefield datasets -- in such cases one can of course use the regular two-image FRC (we do not want to remove anything from it). However in time-lapse image sets of live cells it's rather improbable that the two images in a sequence are the same, assuming that the frame rate is not very high, and thus two-image FRC probably would not work. Also in confocal and STED, single images are rather common. FRC is very sensitive to movements of all sorts, as well as alignment errors, photobleaching, saturation artefacts etc. One-image FRC makes it possible to moderate those issues. More importantly, in image restoration applications, which was the focus here, using two-images would just uselessly double the workload, as two separate images are necessary throughout the image processing workflow.

Comment 16:

The authors may want to read: "High-resolution restoration of 3D structures from widefield images with extreme low signal-to-noise ratio," Muthuvel Arigovindan, Jennifer C. Fung, Daniel Elnatan, Vito Mennella, Yee-Hung Mark Chan, Michael Pollard, Eric Branzlund, John W. Sedat, and David A. Agard, PNAS, 2013, Oct 22, 110(43), 17344-9.

In that article, Arigovindan et al. indeed use FRC to compare different deconvolution algorithms. The authors here could also look at the deconvolution results in that paper which are generally more impressive than the results presented here. Perhaps if the authors here acquire images at lower SNR and show the results of their deconvolution using FRC to get a PSF, then there could be equally impressive results. However, I recognize that that may require taking new data and that might not be feasible.

Answer:

We added the paper to the citations, and we also added a couple examples to the manuscript in which we use regularization with the RL algorithm and show that our FRC based method -- both PSF estimation and progress observation, is compatible with that. In the end regularization (smoothness constraint) is what Arigovindan et al. propose as well, specifically tuned for 3D widefield images. In related projects (we have used this method rather extensively in-house), we have seen the FRC based deconvolution to work rather well with low SNR images as well. Please also see our answer to *Comment 28*.

Comment 17:

The authors claim that the PSF estimated from FRC is "actually rather close to the optimal, as otherwise either blurring or strong increase in noise should be expected." □ Could the authors use another method of finding size of the PSF, for example by measuring line profile across a microtubule and deconvolving that with the known width of microtubules? Alternatively, the authors could acquire a z-stack using sub-resolution-sized beads to acquire a PSF. There are multiple ways the authors could back up their claim that FRC is giving them an accurate PSF measurement.

Answer:

We now compare the one-image FRC measurements throughout the paper with two-image FRC measurements at 1/7 threshold, which has been shown to have excellent correlation with line profile measures in (Nieuwenhuizen, R. P. J. *et al.* Measuring image resolution in optical nanoscopy. *Nat. Methods* **10**, 557 (2013); Tortarolo, G., Castello, M., Diaspro, A., Koho, S. & Vicidomini, G. Evaluating image resolution in stimulated emission depletion microscopy. *Optica* **5**, 32 (2018).). We also investigate the effect of PSF size to RL deconvolution in (Figure S.9).

Comment 18:

If possible, I wonder if FRC could be used to see how the PSF varies with defocus. Could the authors acquire slightly out-of-focus images and, from FRC, see how the width of the PSF varies with defocus. Perhaps see how that PSF width varies with positive or negative defocus as some amount of spherical aberration is likely present. That could be a compelling result.

Answer:

Please refer to (Figure S. 15). Variation of the FRC value with defocus is exactly what we show there.

Comment 19:

Explain rationale for using one-bit as a resolution threshold (for eqn 2). I didn't quite follow the explanation given at around line 276.

Answer:

We now use two different resolution thresholds: $\text{SNR}_e = 0.25$ in FRC, and $\text{SNR}_e = 0.5$ in SFSC. The threshold levels were defined experimentally to closely match two-image FRC values at the 1/7 threshold. The previous explanation was overly complicated, and in the end valid for only certain types of images (low SNR, sparse sampling), so we removed it completely.

Comment 20:

Nieuwenhuizen et al. (whom the authors here cite) found that, using Fourier correlation, the measured resolution was different in directions parallel to versus orthogonal to cellular filaments. In this paper, Koho et al. write that an advantage of FRC/FSC is that it gives "image-independent" results (line 22). Could the authors explain what they mean by "image-independent" and whether that contradicts what others have found regarding a dependence of FRC on the structures imaged?

Answer:

What we wanted to point out in here is that the FRC/FSC measures are objective and scale independent. With images with strong spatial bias in terms of orientation of the structures, the FRC values may be affected as well. In the revised manuscript we in fact discuss this in detail (Figure S.1, Figure S.5). We chose to deal with that issue by taking an average of the two diagonally split sub-image pairs. In extreme cases, one could only measure the resolution in direction orthogonal to the filaments -- our MIPLIB software has a specific iterator for doing that, in which you can choose a section size and orientation (similar to the SFSC idea).

Comment 21:

More on this point of different results for FRC parallel to or orthogonal to imaged filaments. The image used in Figures 3 and 4 is of a microtubulin stained cell. The filaments seem to mostly run at an approx. 45 degree angle from top right to bottom left. The way the image is split to do the FRC uses pixels along an orthogonal line (the pixel in the upper left and the pixel in the lower right of the 2x2 group are used). Could the authors try using the lower left and upper right pixels instead?

Continuing on this point of trying different image splitting schemes. The authors try diagonal splitting ((a) in Supplemental Note 5). They also try diagonal summing ((d) in Supplemental Note 5). It seems to me the authors might want to try doing the diagonal splitting (a), but do so twice (once in each diagonal direction). They could then average the FRC results if the two directions are similar. Or if the results from the different diagonal splittings are different, the authors should explain why that could be.

Answer:

Thank you for an excellent comment. It led us to change the single image based FRC a little bit. The HeLa cell image indeed has a very strongly asymmetric power spectrum (Figure S.1), because the filaments are mostly oriented in a single direction. We deal with this issue by taking the average of the two diagonals as the FRC value, which is more or less the same that normal FRC would do (in fact the one and two image FRC values match). One could of course envision using the higher value instead. As shown in (Figure S.5,) in absence of such bias, the two diagonals match exactly (which is why we didn't notice it before).

Comment 22:

In lines 347-349, the authors state that a blind deconvolution algorithm can be used in which case both the object and the PSF are estimated iteratively. Is that what the authors here are doing? That is, are the authors using FRC to first estimate the PSF and then using a blind deconvolution algorithm to improve on the PSF estimate as well as on the imaged object? From Supplemental Note 3, it seems that the authors are not updating the estimate of the PSF in their deconvolution.

Answer:

In the original manuscript we only estimated the PSF once, at the beginning of the deconvolution. Motivated by Reviewers comment, in the revised manuscript we also run the RL deconvolution by updating the PSF at every iteration. While that seems to work ok, the method without the PSF update appears to actually handle noise a little bit better, and quantitatively the results are pretty much the same. The PSF update scheme appears to lean a little bit towards overfitting, which is often a problem with blind algorithms; we also show how that can, to an extent, be controlled with regularization.

Comment 23:

What are the sizes of the PSF used in the deconvolutions. In Supplementary Figure 5, deconvolution is done with FRC based PSF and a theoretical PSF. What are the widths of those? It would be good to know. It would also help to know how sensitive the deconvolution the authors are using is to the width of the given PSF. Would the results change much if a different FRC threshold were used (such as 1/7 instead of the current threshold)?

Answer: We now added the FWHM information to the manuscript. 200nm based on FRC, 170nm theoretical. As shown in (Figure S.9) deconvolution with a smaller PSF does still work rather well. In (Figure S.6) on the other hand we use a PSF (200nm) that according to the two-image FRC should be larger (220nm), due to the slight overestimation effect that we saw at small pixel sizes -- yet the deconvolution works exquisitely. There is thus some wiggle room in terms of PSF size estimation as well.

Comment 24

In regards to dealing with three-dimensional images with anisotropic resolution. Nieuwenhuizen et al. (2013) use Fourier line correlation (FLC) or Fourier plane correlation (FPC) to deal with anisotropic image resolution. How do those methods compare to the sectioned FSC the authors employ?

Answer:

As was already discussed before, we had somehow misinterpreted the Nieuwenhuizen et al. (2013) 3D implementation. We now implemented the FPC in our MIPLIB software as well, and ran comparisons.

Comment 25

What was the image size used for FRC/SFRC analysis (in pixels)? Could the authors use their methods to find the resolution in different areas of an image? For instance, they could use their methods on 128x128 regions and see how the resolution varies.

PSF estimation. Would the authors be able to measure PSF from FRC on different subregions of the image? This might back up the claims made in the section starting on line 325. Could they show that the PSF is different in regions where the cell is thicker in comparison to where the cell is thinner (supporting idea that cell introduces aberrations). See, for example, Supplemental Figure 12 in Nieuwenhuizen et al. (2013).

Answer:

We now added information of the image sizes to the Image Descriptions. We did not use any specific image size for FRC, but rather used the entire images, or then cropped to focus on some specific region of interest. FRC does not appear to be sensitive at all to the pixel number; our use of the smoothed splines/polynomial fitting might have to do with that, as the curves do not become noisy.

With regards to the local resolution estimation, it is of course possible. We added a figure for that (Figure S. 14). In fact using our MIPLIB software that is rather easy -- here's just a snippet of what is happening under the hood (division to 4 blocks in this case, any pattern is of course possible).

```
1 import itertools
2
3 block_size = list(int(i/2) for i in image1.shape)
4
5 # Get blocks
6 iterables = (xrange(0, m, n) for m, n in zip(image1.shape, block_size))
7 images = []
8 for idx in itertools.product(*iterables):
9     subset_idx = tuple(slice(j, j+k) for j, k in zip(idx, block_size))
10    index = np.array(idx, dtype=int)
11
12    images.append(Image(image1[subset_idx], image1.spacing))
13
14 # Calculate FRC for all
15 results = FourierCorrelationDataCollection()
16
17 for idx in range(len(images)):
18     results[idx] = frc.calculate_single_image_frc(images[idx], args)
19
20
```

PSF estimation in our case equals local resolution estimation, as the former is based on the latter.

Comment 26:

In Figure 6, it would be nice if the tau and eta parameters were defined. Units should be given in the resolution plots (units for resolution missing in other plots too).

Answer:

We now moved those all the way to the Supplementary Material. The τ_1 and η_k are defined in the Methods section. We also cite the original works.

Comment 27:

Supplemental Figure 9. The different labels “Det1_STED_##” mean nothing to the reader. Could the authors explain what these different images are.

Answer:

Please refer to *Comment 10*.

Comment 28

Supplemental Figure 9. Could the authors use their deconvolution methods (with FRC-based PSF) to improve the resolution of, for example, Det2_STED_50? If they could run FRC analysis between the highest resolution image (like Det1_STED_100) and a lower-resolution image before and after the lower-resolution image is deconvolved using FRC-based PSF (similar in spirit to what Arigovindan 2013 did), that would be nice.

Answer:

We now added such FRC plot to every deconvolution example that we show in the manuscript. Here we also show the results for the deconvolution, as suggested by the reviewer -- as it turns out the Det2_STED_50 after blind RL deconvolution is actually better than the raw Det1_STED_100 image. Of course the latter would benefit from deconvolution as well. We did not add this to the manuscript as there in our view is almost too much material already.

Reviewers' comments:

Reviewer #1 (Remarks to the Author):

I am happy to see that Koho et al significantly improved their paper on application of FRC image resolution for image filtering and deconvolution. In particular the ideas about applying FRC in control of convergence in iterative deconvolution algorithms seems to me to be quite useful to the community. The improvements in presentation and thoroughness makes the manuscript publishable. Nevertheless, I have three major objections, where to my mind the evidence does not support the conclusions at all, and that I expect the authors to address:

1. The benchmark of the image splitting procedure for computing a single image based FRC curve compared to state-of-the-art two-image FRC curves, the results of which are shown in Figure 1, can only lead to one conclusion. Namely, qualitatively it is correct, but quantitatively it fails. This is quite apparent from Fig. 1b, but can also be seen in Figs. 1c and d. In Fig. 1b there is a significant effect of over/under-sampling visible for single-image FRC, but hardly for two-image FRC, as is to be expected in view of the sub-sampling that is done to arrive at the single-image FRC curves. I wish to add here that the lack of quantitative agreement does not stand in the way of the envisioned application in convergence control in iterative deconvolution.

2. The use of different thresholds (the 1/7 fixed for two-image FRC and SNR=0.25 curve for single-image FRC) is not motivated at all, only seemingly by the desire to arrive at similar resolution numbers. This is not acceptable. Either the same threshold should be used for both methods, or a physical reason should be given why one threshold is better suited for one method but not for the other.

3. The assessment of FPC and the analysis of it in Figure S13 is sketchy and inconclusive. The curve in S13 bottom row middle suggests a form of low-pass filtering is applied to both image halves, leading to correlations between nothing and nothing and an increase in the curve for high spatial frequencies above the threshold. This suggests there may be problems with the implementation of the method. Other than that, if I look at the polar plots in the top row, who is to say that the SFSC graph is correct but the FPC graph isn't? The authors should do a more thorough assessment or moderate their conclusions.

Reviewer #2 (Remarks to the Author):

The reviewer thanks the author for addressing most of the concerns about the manuscript and the applicability of single image FRC. The manuscript significantly improved and the additional material provided helps in demonstrating the method applicability. Some of the reviewers concerns however have been only partially addressed. Specifically:

1) To prove the generality of the method I find the addition of the Supplementary Note S. 1. of cardinal importance for setting the bases for single image FRC. The implications of the reported calculation are however, in my view, not fully explored. In particular the resulting FRC curve is not compared to the one which would have been obtained with the original image. As an example: following the classical approach, supposing the original image is I than $I_1 = IIIT * I$ and $I_2 = IIIT(x+1) * I$, where $IIIT$ is the Dirac comb function and the FT is simply the convolution with the comb function...how does affect the subsequent calculations? Importantly which are the assumptions on I and hence between I_1 and I_2 for the method to work should be specifically stated/derived. Along these lines, the authors could show how does the spectrum compression influence the obtained resolution (not only for the scaling due to the shift). Although this cannot likely be done in the general case it should be addressable for some special cases such as a perfect gaussian object or a double line. I believe this analysis would reveal the application boundaries (e.g. the technique will clearly fail with some unrealistic extreme cases such as a 1px period

checkboard pattern) of the technique in a way complementary to the experimental approach adopted by the authors.

2) While the readability of the manuscript has been highly improved, Fig. 1) is very hard to decipher. I would suggest adding a sub-panel where the explicit obtained resolutions by double and single FRC are plotted in the same graph for panels b), c) and d). Also, for panel c) I could not find the corresponding STED images anywhere in the manuscript or the supplementary material.

3) More as a suggestion, since it bears no direct scientific relevance, I would find very useful for the community, additionally to providing the corresponding libraries as the authors do, a simple one click plugin/stand-alone program to obtain the resolution of a single image with the diagonal splitting method.

Reviewer #3 (Remarks to the Author):

I believe the authors have adequately responded to most of the reviewer comments. The organization of the paper has improved and the methods described more clearly.

However, I still think the organization could be improved. And I think some of the methods and particularly the conclusions could be stated more succinctly and clearly.

1) The first paragraph on page 4 on testing FRC on a single split image with different pixel sizes reads like a list of experiments without a clear point. I think the overall conclusion should be stated succinctly and the details left to the supplementals.

2) The paragraph on page 5 on different sub-sampling methods. Not sure this is necessary. Are there any benefits to using any of the methods? If the other methods suffer from spatial biasing and do not provide bandwidth gains, why bother discussing them?

Other comments:

1) The authors should better explain the $1/7$ threshold (for two-image FRC) and $SNR=0.25$ (for one image). Why is it that there are two thresholds?

2) The discussion on page 4 about Figure 1b. I think the authors should try to arrive at a more solid conclusion. To me, it seems that Figure 1b shows that the one-image FRC method only works if the pixel size is kept between certain values. That ought to be stated clearly and unambiguously in the text. Because this fact limits the applicability of this method.

3) In figure 2, the authors show the result of filtering based on FRC cut-off. I think a necessary image in that figure would be the result of filtering based on just a theoretical estimate of the PSF. The authors are trying to show that with FRC, they can improve the results of frequency domain filtering. So they should show that this is an improvement based on what someone using frequency domain filtering would do if that person didn't have this FRC method and just relied on a theoretical estimate of the PSF size.

4) On page 8, different deconvolution algorithms are discussed. Then, it is stated that "all the four algorithms produce results with nearly identical effective resolution." Why is that? And if that is the case what is the purpose of showing all four?

5) At the bottom of page 8, it is written that "another long-standing problem with RL and other iterative algorithms [is] that one does not really know when the algorithm should be stopped." I think it would help if the authors provide references backing up their claim that this is a problem.

6) In regards to the above comment, if the authors are making a claim that their method allows one to determine how many iterations to use, I think they ought to have a plot of resolution vs iteration in the main text and not in the supplementals. That plot ought to convincingly show that their method provides a robust way to determine how many iterations to use.

7) Order of supplemental figures. Figure S1 legend starts out with "As was shown in Figure S5..." Well, since $5 > 1$, it hasn't in fact been shown. Seems like it ought to be placed after S5.

8) Bottom of page 10, discussion of "zero crossing of the second derivative" as an "alternative stopping point." I think this needs more explanation or justification. Just showing one example where this might apply doesn't seem that convincing.

9) Top of page 11. The authors say that turning diagonal averaging off increases noise sensitivity. Could the authors explain. I'm not sure I understand why. And if that's the claim, they should

show the same data shown in Fig S10 with the diagonal averaging on.

10) Bottom of page 11, it is stated that the STED image used for SFSC/FPC is a 300x300x300 pixel region. Does it really have 300 pixels in z? It doesn't seem like it.

11) Figure 4, STED image. Could the authors show a zoomed in region? I think the results here are more impressive than the results in Fig 2. So I don't understand why Fig 2 is given so much space in comparison.

Answers to Reviewers

We thank the Referees for their patience, positive feedbacks, as well as for the great comments that helped us improve our manuscript. We have checked all the comments provided by the Referees and have made necessary changes accordingly to their indications.

Reviewer #1:

Comment 1:

The benchmark of the image splitting procedure for computing a single image based FRC curve compared to state-of-the-art two-image FRC curves, the results of which are shown in Figure 1, can only lead to one conclusion. Namely, qualitatively it is correct, but quantitatively it fails. This is quite apparent from Fig. 1b, but can also be seen in Figs. 1c and d. In Fig. 1b there is a significant effect of over/under-sampling visible for single-image FRC, but hardly for two-image FRC, as is to be expected in view of the sub-sampling that is done to arrive at the single-image FRC curves. I wish to add here that the lack of quantitative agreement does not stand in the way of the envisioned application in convergence control in iterative deconvolution.

Answer:

Thank you once again for a great comment that helped us both to improve the one-image FRC method, as well as to in general, to focus our manuscript further to the subject of image restoration and iterative deconvolution – which in fact was our initial intention. We now give a much smaller role to the one-image FRC, and show that the proposed methods are feasible with two-image FRC as well.

As it turns out, the quantitative inaccuracy of the one-image FRC was mostly due to our inappropriate rescaling of the frequency axis. We now took a closer look at the data that we have (specifically the datasets with pixel size gradients) and found a way to produce rather accurate quantitative measures as well with the one-image FRC. The one-image FRC measures are of course bandwidth limited, and ideally two-image measures should also in our opinion be used for quantitative measures, when possible. We now clearly state this point in the manuscript (lines 129-132), together with the reviewer's comment about one-image FRC and convergence control of iterative deconvolution :

“With one-image FRC, quantitative measures at $d_{min}/3.5$ sampling densities should thus be taken with a grain of salt -- or preferably, two-image FRC should be used, if possible. The observed attenuation however does not stand in the way of using one-image FRC in the envisioned application in convergence control in iterative deconvolution.”

Comment 2:

The use of different thresholds (the 1/7 fixed for two-image FRC and SNR=0.25 curve for single-image FRC) is not motivated at all, only seemingly by the desire to arrive at similar resolution numbers. This

is not acceptable. Either the same threshold should be used for both methods, or a physical reason should be given why one threshold is better suited for one method but not for the other.

Answer:

This issue has now been fixed. The 1/7 threshold is used with both one and two image FRC.

Comment 3:

The assessment of FPC and the analysis of it in Figure S13 is sketchy and inconclusive. The curve in S13 bottom row middle suggests a form of low-pass filtering is applied to both image halves, leading to correlations between nothing and nothing and an increase in the curve for high spatial frequencies above the threshold. This suggests there may be problems with the implementation of the method. Other than that, if I look at the polar plots in the top row, who is to say that the SFSC graph is correct but the FPC graph isn't? The authors should do a more thorough assessment or moderate their conclusions.

Answer:

We took a closer look at the FPC results and were indeed able to improve them. We now apply the same correction for the anisotropic sampling that we use with FRC/SFSC and indeed the results were greatly improved. FPC however seems to suffer quite a bit from the various sorts of artefacts that are present in the axial direction that are well visible e.g. on the z-axis of the OTF plots in Fig S. 10 c), which causes it to somewhat underestimate the resolution in the axial resolution. It may be that our correction is not optimal for FPC, but then again except for the $\sim 90^\circ$ orientation it seems to work perfectly. In the SFSC we use particular iterator to exclude the voxels directly on the z-axis to deal with this issue. We are not quite sure about the origin of the artefacts (they are present in the raw data already), but we see similar artefacts almost in every confocal stack. FPC does thus work correctly, but our datasets are just a little bit difficult. We adjusted our conclusions accordingly.

As what comes to the apparent low-pass filtering effect, this is of course intrinsic to the images, as the resolution in the axial direction, in case of the STED image is approx $\frac{1}{4}$ of the lateral resolution – and also the data is more sparsely sampled in the axial direction with respect to the lateral direction. For this reason, before analysis we need to resample the stacks to isotropic pixel size as well as to apply zero padding to create a cubic shape. In FPC similarly the rotation requires some sort of resampling. In practice, in the axial direction the high frequencies of the power spectrum have near zero energy, which creates the correlations of nothing with nothing as the reviewer suggests.

As for which graph is right, we now added a new plot Fig S. 10 c), which illustrates the difference of SFSC and FPC quite clearly.

Reviewer #2:

Comment 4:

The reviewer thanks the author for addressing most of the concerns about the manuscript and the applicability of single image FRC. The manuscript significantly improved and the additional material provided helps in demonstrating the method applicability. Some of the reviewers concerns however have been only partially addressed. Specifically: 1) To prove the generality of the method I find the addition of the Supplementary Note S. 1. of cardinal importance for setting the

bases for single image FRC. The implications of the reported calculation are however, in my view, not fully explored. In particular the resulting FRC curve is not compared to the one which would have been obtained with the original image. As an example: following the classical approach, supposing the original image is I then $I_1 = \text{IIIT} * I$ and $I_2 = \text{IIIT}(x+1) * I$, where IIIT is the Dirac comb function and the FT is simply the convolution with the comb function...how does it affect the subsequent calculations? Importantly which are the assumptions on I and hence between I_1 and I_2 for the method to work should be specifically stated/derived. Along these lines, the authors could show how does the spectrum compression influence the obtained resolution (not only for the scaling due to the shift). Although this cannot likely be done in the general case it should be addressable for some special cases such as a perfect gaussian object or a double line. I believe this analysis would reveal the application boundaries (e.g. the technique will clearly fail with some unrealistic extreme cases such as a 1px period checkboard pattern) of the technique in a way complementary to the experimental approach adopted by the authors.

Answer:

We want to thank the reviewer for a very detailed comment that helped us to improve the one-image FRC method significantly. Regrettably, not for the small part due to our, at best modest command of the sampling theory, we have not for the moment managed to obtain an analytical solution that we would have the courage to show in public. This thus remains a subject for future work.

However, in the revised manuscript we took our experimental approach a step further and managed to find a rescaling for the frequency axis that reflects the modulation effect in the FRC curves. We have no delusions, that the “calibration” curve that we apply, may not be analytically optimal, but it is shown to work rather well across a wide range of sampling densities, and should be more than adequate for the application that we intended for the one-image FRC.

Comment 5:

While the readability of the manuscript has been highly improved, Fig. 1) is very hard to decipher. I would suggest adding a sub-panel where the explicit obtained resolutions by double and single FRC are plotted in the same graph for panels b), c) and d). Also, for panel c) I could not find the corresponding STED images anywhere in the manuscript or the supplementary material.

Answer:

We have now revised almost all the figures, and also rewrote most of the results for better readability and focus. This entailed adding some new results, especially related to the one-image FRC, as well as the background correction in RL deconvolution, and we also removed a lot of supplementary figures that in the end did not add any value to the reader. All images used for the various results are now shown in the manuscript and information regarding the frame/pixel sizes are stated in the supplementary material.

Comment 6:

More as a suggestion, since it bears no direct scientific relevance, I would find very useful for the community, additionally to providing the corresponding libraries as the authors do, a simple one click plugin/stand-alone program to obtain the resolution of a single image with the diagonal splitting method.

Answer:

Yes, there have been some request for a Fiji plugin. We however have very little experience in Fiji development. When time allows, as we see the value in this, we intend to look into the existing FRC plugin in Fiji to see whether it could be easily extended for one-image FRC.

Our efforts are increasingly focusing on Python, as it allows the integration of image processing methods with other powerful tools (e.g. plotting, machine learning etc). We will supply some Jupyter Notebook examples with our MIPLIB to try to make the library easier to use.

Reviewer #3 :

Comment 7:

The first paragraph on page 4 on testing FRC on a single split image with different pixel sizes reads like a list of experiments without a clear point. I think the overall conclusion should be stated succinctly and the details left to the supplementals.

Answer:

We first want to thank the reviewer for excellent comments that were very helpful in finding a better focus for our manuscript. Most of the results section was revised for easier readability. We tried to significantly simplify the one-image FRC explanation, and in fact removed a lot of results that were mainly confusing. We hope that the explanation of the one-image FRC calibration is now more clear and to the point.

Comment 8:

The paragraph on page 5 on different sub-sampling methods. Not sure this is necessary. Are there any benefits to using any of the methods? If the other methods suffer from spatial biasing and do not provide bandwidth gains, why bother discussing them?

Answer:

Thank you, we agree. They are gone now.

Comment 9:

The authors should better explain the 1/7 threshold (for two-image FRC) and SNR=0.25 (for one image). Why is it that there are two thresholds?

Answer:

As it turns out with the SNR=0.25 scaling we were in a way compensating for the non-linearity of the frequency axis. In the revised manuscript, the one-image FRC now works more logically, and the same 1/7 threshold is used in both one and two image FRC

Comment 10:

The discussion on page 4 about Figure 1b. I think the authors should try to arrive at a more solid conclusion. To me, it seems that Figure 1b shows that the one-image FRC method only works if the pixel

size is kept between certain values. That ought to be stated clearly and unambiguously in the text. Because this fact limits the applicability of this method.

Answer:

As was already explained earlier, the strange behavior of the one-image FRC was due to inappropriate scaling of the frequency axis. Now, as our results show, one image FRC works rather well up to circa $d_{\min}/3.5$ sampling density, after which it clearly starts to attenuate.

Comment 11:

In figure 2, the authors show the result of filtering based on FRC cut-off. I think a necessary image in that figure would be the result of filtering based on just a theoretical estimate of the PSF. The authors are trying to show that with FRC, they can improve the results of frequency domain filtering. So they should show that this is an improvement based on what someone using frequency domain filtering would do if that person didn't have this FRC method and just relied on a theoretical estimate of the PSF size.

Answer:

The FFT filtering part was now greatly simplified, and integrated into Figure 1. We also added a new Figure S. 1, in which we compare the different filtering methods, also based on the theoretical threshold, as proposed by the reviewer.

Comment 12:

On page 8, different deconvolution algorithms are discussed. Then, it is stated that “all the four algorithms produce results with nearly identical effective resolution.” Why is that? And if that is the case what is the purpose of showing all four?

Answer:

We wanted to show the four algorithms mainly for two reasons:

1° to show different ways in which FRC can be leveraged in iterative deconvolution and to compare their effectiveness

2° for “educational” purpose, to show the reader the subjective differences of the results obtained with the various algorithms. While the resolution in all of the methods is nearly the same, the end results look very different. We now changed the image Figure 3 to highlight these differences.

In all the other examples Fig. S5 – Fig. S8, the regular RL algorithm with fixed PSF is now consistently used, as it produces the best results in Figure 3.

Comment 13:

At the bottom of page 8, it is written that “another long-standing problem with RL and other iterative algorithms [is] that one does not really know when the algorithm should be stopped.” I think it would help if the authors provide references backing up their claim that this is a problem.

Answer:

We added a couple of references. It is a quite commonly known problem, yet the literature on the subject, especially related to microscopy image processing, appears to be rather scarce. The τ_1 measure that we show in the manuscript, appears to be in a rather common use.

Comment 14:

In regards to the above comment, if the authors are making a claim that their method allows one to determine how many iterations to use, I think they ought to have a plot of resolution vs iteration in the main text and not in the supplementals. That plot ought to convincingly show that their method provides a robust way to determine how many iterations to use.

Answer:

Such plots are now visible in all figures dealing with iterative deconvolution. We propose two practical stopping conditions, based on the Δ Resolution curve: 1nm/it and 0.2nm/it. Their motivation is explained in the text and illustrated with examples. It is possible also to let the deconvolution converge to the maximum resolution, but that may not make practical sense in most cases, as after e.g. the 0.2nm/it threshold, very little is gained in terms of image quality.

Comment 15:

Order of supplemental figures. Figure S1 legend starts out with “As was shown in Figure S5...” Well, since $5 > 1$, it hasn’t in fact been shown. Seems like it ought to be placed after S5.

Answer:

That was fixed.

Comment 16:

Bottom of page 10, discussion of “zero crossing of the second derivative” as an “alternative stopping point.” I think this needs more explanation or justification. Just showing one example where this might apply doesn’t seem that convincing.

Answer:

We now revised the discussion regarding the behavior of FRC in images with strong background. The simulation, hopefully better explained now, is also complemented with a real example in Figure S. 8. In addition to characterizing the background peak in the FRC, and showing how the Δ^2 Resolution curve works, we now also suggest a method for removing the background in the blind RL deconvolution.

Comment 17:

Top of page 11. The authors say that turning diagonal averaging off increases noise sensitivity. Could the authors explain. I’m not sure I understanding why. And if that’s the claim, they should show the same data shown in Fig S10 with the diagonal averaging on.

Answer:

We now removed the whole discussion about turning the diagonal averaging off, as we wanted to un-complicate things a little bit. The averaging is always kept on.

Comment 18:

Bottom of page 11, it is stated that the STED image used for SFSC/FPC is a 300x300x300 pixel region. Does it really have 300 pixels in z? It doesn't seem like it.

Answer:

Indeed, that was a little bit misleading. The image block size is 300x300x30 voxels, but for the 3D analysis one of course needs a cubic shape. Therefore, resampling in Z (to make cubic voxels) and zero padding is used to create a shape of 300x300x300. We now mention the 300x300x30 in the main text.

Comment 19:

Figure 4, STED image. Could the authors show a zoomed in region? I think the results here are more impressive than the results in Fig 2. So I don't understand why Fig 2 is given so much space in comparison.

Answer:

We agree. Fig 2 was now altogether removed, as the FFT filtering is just a very small part of the manuscript.

Reviewers' comments:

Reviewer #1 (Remarks to the Author):

The authors have addressed the concerns I have raised, I think the paper is publishable now.

Reviewer #2 (Remarks to the Author):

The manuscript significantly improved compared to the previous submission both in term of scientific accuracy and readability and it is now potentially suitable for publication. However few main concern still needs to be addressed before publication:

* The calibration for the rescaling of the frequency axis is shown to correct the discrepancies between single and double image FRC across different sampling ratio. It is however not entirely clear the logic for choosing this specific shape for the rescaling. Even if not "analytically optimal" the author should still justify their choice of this specific functional dependency of the rescaling as an approximation of an exact correction.

* The treshold of 1nm/iteration (or 2) for the stopping criterion in the resolution gradient is somehow arbitrary and the author should change it with a relative improvement (or any other universally applicable measure).

Minor misspelling:

* At line 229 the authors wrote "microtubulin" stained. Either microtubules or tubulin.

* In line 474 and 476 the authors write "nominator". It should be changed with numerator.

Reviewer #3 (Remarks to the Author):

The authors have significantly improved the paper and adjusted the focus away from single image FRC. I point out a few areas for improvement but I do not think any major issues remain in the way of publishing.

1) The issue of rescaling the frequency axis for the one image FRC (starting around line 114) was a bit confusing. There were some grammatical issues in that description.

2) Blink RL deconvolution. The authors use three algorithms: regular, adjusted, adjusted total variation regularization. It might be nice if the authors go into some details about what to expect. Is it expected, given the assumptions that go into / the principles of RL deconvolution that updating the PSF would improve results or not?

3) Two-image FRC used to evaluate RL deconvolution in Figure S8. The authors write, beginning around line 77, about how two-image FRC for tracking the progress of deconvolution may involve "complex software implementation, a significant computational overhead...". Since the authors *have* done this perhaps they can be more specific about the complexity / hardware resources.

Smaller things:

1) Line 49: I think it should be: "algorithms that leverage FRC ..."

2) Caption of Figure 2. Currently reads "First and image is split into" (and -> the ?)

3) Line 133: "Having established, how..." -> remove comma

4) Line 204/5: "method for estimating background introduced..." Word "is" missing?

Answers to Reviewers

We thank the Referees for their patience, positive feedbacks, as well as for the great comments that helped us improve our manuscript. We have checked all the comments provided by the Referees and have made necessary changes accordingly to their indications.

Reviewer #1 (Remarks to the Author):

The authors have addressed the concerns I have raised, I think the paper is publishable now.

We want thank the Reviewer #1 for excellent comments throughout the review process that helped us to significantly improve the quality of the manuscript..

Reviewer #2 (Remarks to the Author):

The manuscript significantly improved compared to the previous submission both in term of scientific accuracy and readability and it is now potentially suitable for publication.

We want thank the Reviewer #2 for excellent comments throughout the review process that helped us to significantly improve the quality of the manuscript.

However few main concern still needs to be addressed before publication:

* The calibration for the rescaling of the frequency axis is shown to correct the discrepancies between single and double image FRC across different sampling ratio. It is however not entirely clear the logic for choosing this specific shape for the rescaling. Even if not "analytically optimal" the author should still justify their choice of this specific functional dependency of the rescaling as an approximation of an exact correction.

Thank you for an excellent comment that helped us to spot an important error in our one-image FRC implementation.

We now did a proper calibration of the one-image FRC and added a new supplementary note (Note S. 2) in which we explain how it is done. A new Supplementary figure (Figure S. 3) was added as well for the calibration results. It is now shown exactly how the calibration curve is obtained. As a consequence of the added material, we were able to simplify the one-image FRC description and Figure 2 in the main text significantly.

Moreover, as it turns out, the logarithmic curve was indeed wrong, and we also found a bug in our one-image FRC software, which made it necessary to update the deconvolution results in which we used one-image FRC. This does not change our conclusions in any way, but the numerical values are somewhat different.

* The treshold of 1nm/iteration (or 2) for the stopping criterion in the resolution gradient is somehow arbitrary and the author should change it with a relative improvement (or any other universally applicable measure).

We added the following to the text (lines 150-160) to express the thresholds on a relative scale

The specified resolution thresholds assume a nanometer length scale, which is meaningful in optical microscopy. One of course needs to adjust for appropriate physical units, when working with images, with significantly higher or lower resolution. The 1nm/it threshold corresponds to roughly 0.5 per-cent and 0.02nm/it to 1 permille of the resolution.

Minor misspelling:

* At line 229 the authors wrote "microtubulin" stained. Either microtubules or tubulin.

Fixed

* In line 474 and 476 the authors write "nominator". It should be changed with numerator.

Fixed

Reviewer #3 (Remarks to the Author):

The authors have significantly improved the paper and adjusted the focus away from single image FRC. I point out a few areas for improvement but I do not think any major issues remain in the way of publishing.

We want thank the Reviewer #3 for excellent comments throughout the review process that helped us to significantly improve the quality of the manuscript.

1) The issue of rescaling the frequency axis for the one image FRC (starting around line 114) was a bit confusing. There were some grammatical issues in that description.

As already discussed above, we now significantly simplified the description of the one-image FRC calibration and also added new material in the Supplementary Note, in which we explain in detail how the rescaling is done.

2) Blink RL deconvolution. The authors use three algorithms: regular, adjusted, adjusted total variation regularization. It might be nice if the authors go into some details about what to expect. Is it expected, given the assumptions that go into / the principles of RL deconvolution that updating the PSF would improve results or not?

We made the description of the RL algorithms a bit more conversational, and added more details about why one might be interested in updating the PSF during iteration. Updating the PSF may become necessary, if the initial estimate is very far from an optimal one, e.g. with very low-quality images.

As a further note, in the paper we mainly wanted to demonstrate that implementing FRC based PSF estimation with all three algorithms is feasible. There are no universal solutions in image processing, and thus in our view it's important to show what can be done, if necessary.

3) Two-image FRC used to evaluate RL deconvolution in Figure S8. The authors write, beginning around line 77, about how two-image FRC for tracking the progress of deconvolution may involve "complex software implementation, a significant computational overhead...". Since the authors *have* done this perhaps they can be more specific about the complexity / hardware resources.

The deconvolution was run separately with the two images and then the FRC was calculated afterwards, for each intermediate result image pair.

We accidentally deleted that crucial sentence at some editing stage of our manuscript revision. We added that to the figure caption now. We did not thus implement the synchronised two-image deconvolution. It would, of course be possible to implement such an algorithm, and on a programming language supporting proper multi-threading it might also be relatively efficient. We wanted to keep things simple here, and make the method that we propose easy for anyone to implement — that is the point that we wanted to make.

Smaller things:

1) Line 49: I think it should be: “algorithms that leverage FRC ...”

Fixed

2) Caption of Figure 2. Currently reads “First and image is split into” (and -> the ?)

Fixed

3) Line 133: “Having established, how...” -> remove comma

Fixed

4) Line 204/5: “method for estimating background introduced...” Word “is” missing?

Fixed

REVIEWERS' COMMENTS:

Reviewer #2 (Remarks to the Author):

With the inclusion of the proper description of the calibration procedure the authors have properly addressed my remaining concerns and the manuscript is now suitable for publication.